# Immunosuppression causes dynamic changes in expression QTLs in psoriatic skin

Qian Xiao[1,2,3,4,11], Joseph Mears[1,2,3,4,11], Aparna Nathan[1,2,3,4,5], Kazuyoshi Ishigaki [1,2,3,4,6], Yuriy Baglaenko[1,2,3,4], Noha Lim[7], Laura A. Cooney[7,8], Kristina M. Harris[7], Mark S. Anderson [7], David A. Fox[8], Dawn E. Smilek[7], James G. Krueger[9] & Soumya Raychaudhuri [1,2,3,4,5,10] ✉

Psoriasis is a chronic, systemic inflammatory condition primarily affecting skin. While the role of the immune compartment (e.g., T cells) is well established, the changes in the skin compartment are more poorly understood. Using longitudinal skin biopsies ($n = 375$) from the "Psoriasis Treatment with Abatacept and Ustekinumab: A Study of Efficacy"(PAUSE) clinical trial ($n = 101$), we report 953 expression quantitative trait loci (eQTLs). Of those, 116 eQTLs have effect sizes that were modulated by local skin inflammation (eQTL interactions). By examining these eQTL genes (eGenes), we find that most are expressed in the skin tissue compartment, and a subset overlap with the NRF2 pathway. Indeed, the strongest eQTL interaction signal – rs1491377616-*LCE3C* – links a psoriasis risk locus with a gene specifically expressed in the epidermis. This eQTL study highlights the potential to use biospecimens from clinical trials to discover in vivo eQTL interactions with therapeutically relevant environmental variables.

Psoriasis is a chronic inflammatory systemic disorder primarily involving skin that affects approximately 125 million people across the world[1]. Genetic predisposition, environmental factors, and immune dysregulation all play a critical role in disease development[2]. Psoriasis can cause thickened red skin lesions characterized by keratinocyte hyperproliferation, angiogenesis, and immune cell infiltration[3,4]. In about 30% of cases, psoriasis is associated with an inflammatory arthritis[5]. Dermal and epidermal cell changes in psoriatic lesions are the consequence of a pathogenic immune response in the skin. As current knowledge suggests[6], activated dendritic cells and IL-17-producing T cells trigger an inflammatory cascade, promoting downstream keratinocyte proliferation and other dermatological disease

manifestations[4,7–10]. The clinical efficacy of immunomodulatory cytokine antagonists, such as ustekinumab (anti-IL-12/IL-23) and adalimumab (anti-TNF), confirms the key function of the immune system in psoriasis[11–13].

However, the dermal and epidermal compartments are increasingly recognized as an important part of disease initiation and sustained chronic inflammation through induction of the innate immune response and recruitment of inflammatory immune cells[14,15]. For instance, in the early events of psoriasis, keratinocytes are a key source of innate immune mediators such as antimicrobial peptides that can activate plasmacytoid and myeloid dendritic cells, which can in turn initiate subsequent adaptive immune responses[14]. Understanding the

[1]Center for Data Sciences, Brigham and Women's Hospital, Boston, MA, USA. [2]Division of Genetics, Department of Medicine, Brigham and Women's Hospital, Boston, MA, USA. [3]Division of Rheumatology, Inflammation, and Immunity, Department of Medicine, Brigham and Women's Hospital and Harvard Medical School, Boston, MA, USA. [4]Program in Medical and Population Genetics, Broad Institute of MIT and Harvard, Cambridge, MA, USA. [5]Department of Biomedical Informatics, Harvard Medical School, Boston, MA, USA. [6]Laboratory for Human Immunogenetics, RIKEN Center for Integrative Medical Sciences, Yokohama City, Kanagawa, Japan. [7]Immune Tolerance Network, Diabetes Center, University of California, San Francisco, San Francisco, CA, USA. [8]Division of Rheumatology, Department of Internal Medicine and Clinical Autoimmunity Center of Excellence, University of Michigan, Ann Arbor, MI, USA. [9]Laboratory for Investigative Dermatology, The Rockefeller University, New York, NY, USA. [10]Centre for Genetics and Genomics Versus Arthritis, Centre for Musculoskeletal Research, The University of Manchester, Manchester, UK. [11]These authors contributed equally: Qian Xiao, Joseph Mears. ✉e-mail: soumya@broadinstitute.org

dynamics of dermal and epidermal cells in the context of psoriasis may open unexplored avenues for novel therapeutics targeting inflammation initiation.

Genome-wide association studies (GWAS) have characterized the genetic architecture of psoriasis in diverse populations, and identified risk loci related to the immune system, such as Major Histocompatibility Complex (MHC) and NF-kappa B pathways[16–21]. However, the majority of psoriasis-associated loci reside in non-coding regions, like functional enhancers, that may regulate expression of nearby genes[22]. One way to understand gene regulation in psoriasis tissue is to examine expression quantitative loci (eQTLs), genetic variants (i.e., eSNPs) that alter the expression level of a gene (i.e., eGene) located nearby (*cis*-eQTL) or distant (*trans*-eQTL). eQTLs can be shared between cell types or cell states, or can be context-dependent[23]. For instance, an environmental factor may alter regulatory factors that bind the eSNP, resulting in a genotype-by-environment statistical interaction, where the eQTL effect is amplified or dampened by the presence of the environmental factor[24–29]. In the context of psoriasis, inflammatory status of the skin may alter regulatory factors that bind the eSNP, leading to different degrees of eQTL gene expression changes upon the same eQTL. Here, eQTL interactions may serve as a proxy for regulatory changes in the immune or dermal and epidermal cell states when the status of the skin shifts from its non-lesional state to a psoriatic lesion.

Other approaches such as differential expression or single-cell genomics may also be deployed to investigate molecular changes in psoriasis. Single-cell studies have reported enrichment of IL-17A-expressing T cells and antimicrobial peptide-expressing differentiated keratinocytes in lesional psoriatic skin[30–32]. Just as single-cell data may be a useful strategy to capture differences in cell states; analytical approaches such as pseudotime analysis and differential abundance analysis[33–35] help infer the progression of cells through biological processes. However, these approaches may not reveal systematic changes in gene regulation and the potentially pathogenic variants mediating these changes. For example, differentially expressed genes or differentially abundant cell states suggest downstream functional consequences or disease characteristics, but eQTL interactions can point to shared upstream regulatory mechanisms by prioritizing TFs with relevant binding motifs, and identify potential causal variants and genes involved in complex diseases[36]. Therefore, eQTL interactions may complement other approaches.

Here we analyzed genotype and tissue transcriptional data from 77 participants with psoriasis from Immune Tolerance Network (ITN) Psoriasis Treatment with Abatacept and Ustekinumab: A Study of Efficacy (PAUSE) trial[37]. The trial was conducted to determine whether costimulatory blockade with abatacept could prevent psoriasis relapse after withdrawal of ustekinumab, an FDA-approved treatment for psoriasis. In this trial abatacept did not successfully suppress the pathogenic psoriasis transcriptional signature in skin after ustekinumab withdrawal, nor did it prevent clinical psoriasis relapse[37].

In observational studies, disease-specific effects can be hard to disentangle from the effect of drugs that the patients may be using[38]. Often since patients with more aggressive disease are treated with more aggressive therapies, separating the effect of the medications and disease is challenging. Randomized clinical trials are designed for valid comparison between two or more groups of subjects that, due to randomization, are well-controlled for therapeutically important variables. Moreover, the PAUSE trial collected genotyping data, along with repeat measurements of clinical psoriasis response metrics and skin biopsy transcriptional profiles. In this first eQTL study of psoriasis using data from a clinical trial, longitudinal skin biopsies from participants allows assessment of *cis*-eQTL associations between genetic variants and expression levels of nearby genes. By leveraging the power of repeat clinical measurements, we also explore how eQTLs are modified by the inflammatory status of the skin or other clinical variables. These eQTL interactions can inform our understanding of the regulatory mechanisms and immune, dermal, and epidermal cell involvement in the disease.

## Results

### Mapping eQTLs in patients with psoriasis

We obtained longitudinal lesional and non-lesional skin biopsies from participants at baseline, during treatment, and at the time of psoriasis relapse after study medication withdrawal over a course of 22 months. We used genome-wide genotyping and RNA-seq to assay samples. After stringent quality control, we had RNA-seq data on 375 samples from 77 genotyped patients (Fig. 1a, b, Supplementary Table 1, and Supplementary Data 1). We had genotyping data on 731,068 SNPs from the Illumina Infinium Multi-Ethnic Global BeadChip. After imputation, we had a total of 2,074,125 SNPs with imputation score $r^2 > 0.99$ and MAF > 0.05.

First, we identified cis-eQTLs using all skin biopsies. To ensure robust expression measurements, we restricted our analysis to 27,100 well-expressed genes (>0.1 TPM and counts >6 across for >20% of samples). To detect *cis*-sQTLs, we queried SNPs within 250 kb of the TSS of each gene, as recommended in previous studies[36,39,40]. Since individuals had both longitudinal lesional and non-lesional skin biopsies, we used a linear mixed effects model to test the effect of SNP genotype on gene expression while accounting for repeat expression measurements from the same individual with a random effect ("Methods"). We included 20 gene expression PCs to control for genotype-independent differences in global expression between cell states, 3 genotyping PCs to control for ancestry, and recruitment site as covariates. We included principal components as covariates in our model to account for confounding sources of gene expression variation that are not limited to those that have been measured in the study. We chose 20 PCs to maximize the number of eQTL genes detected while minimizing the number of principal components that explain a reasonable amount of variance (82.7% variance, Supplementary Fig. 1a). We observed that the number of eQTLs identified was not very sensitive to the choice of $n = 20$ PCs, with a similar number of eQTLs being identified using 15-25 PCs (Supplementary Fig. 1b).

Of 6,305,752 SNP–gene pairs tested, using a stringent Bonferroni $p$ value threshold ($p < 6.69e{-}9 = 0.05/6,305,752$), we reported 24,374 significant SNP–gene pairs (Fig. 1c). With only the lead SNP for each gene, we identified 953 genes with at least one significant eQTL (eGenes) (Supplementary Data 2).

As a complementary analysis, we reran the analyses without accounting for multiple visits to see how the current model compares to a linear model using only the baseline samples ($N = 140$ samples versus 375 samples). For the first visit analysis we included 140 samples (generally one lesional and one non-lesional sample) from 74 individuals. For the analysis with all visits we included 375 samples from 77 individuals. As expected, linear mixed model including all the visits detected more eQTLs than the first time-point linear model ($N = 953$ vs. $N = 575$). While the effect sizes of the two models are highly concordant comparing significant eQTL pairs identified using the full model ($R = 1$), the $t$-values from the linear mixed model are consistently higher than the linear model (Supplementary Fig. 2a, b), indicating that including repeat visits increases the power of detecting eQTL signals.

To confirm that our eQTLs were consistent with prior eQTLs reported in the skin, we compared significant SNP–gene pairs from this dataset to *cis*-eQTLs reported by the GTEx consortium among 517 healthy skin samples from 449 individuals who were not selected for a particular condition[28]. We examined significant eQTLs from GTEx and the pair of lead SNPs and eGenes for each of these eQTLs ($N = 2621$ eGenes). We observed a high degree of concordance between t-values of deduplicated SNP–gene pairs the PAUSE study and GTEx, suggesting our study identified highly concordant results with a study that is substantially larger in size (with 98.5% of effects in the same direction,

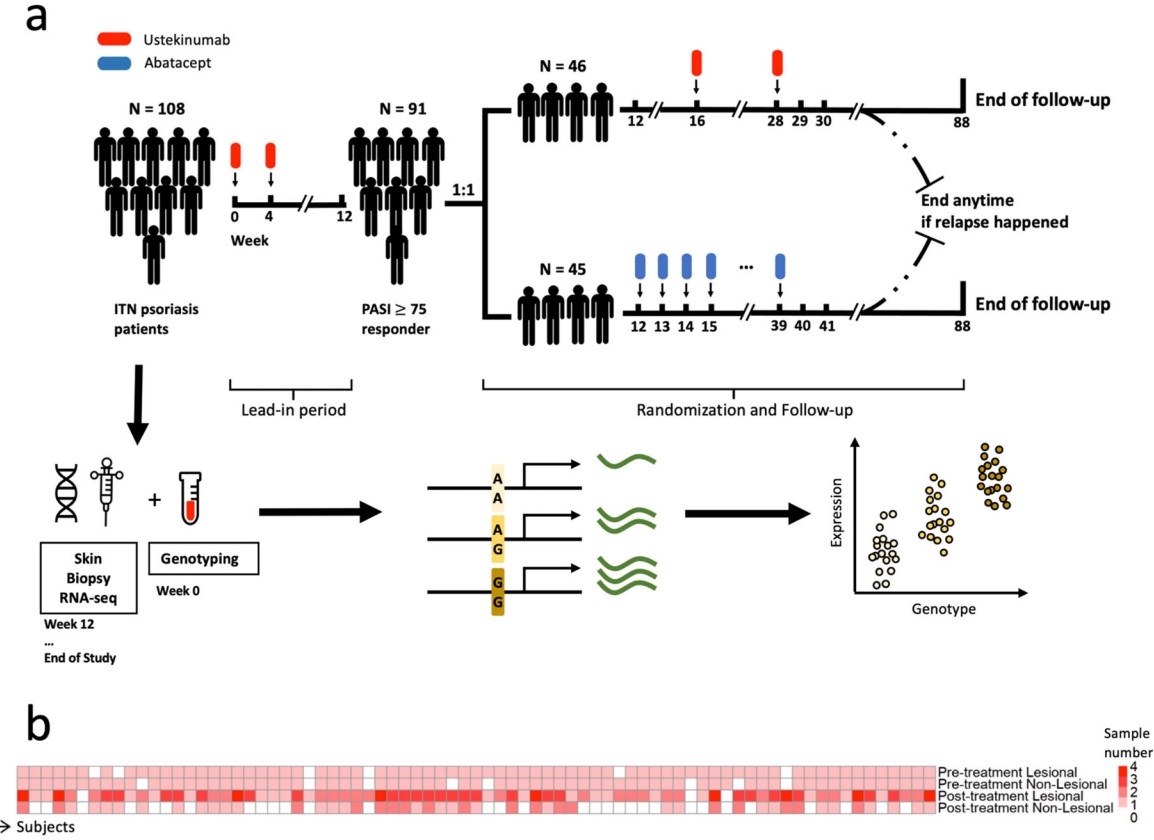

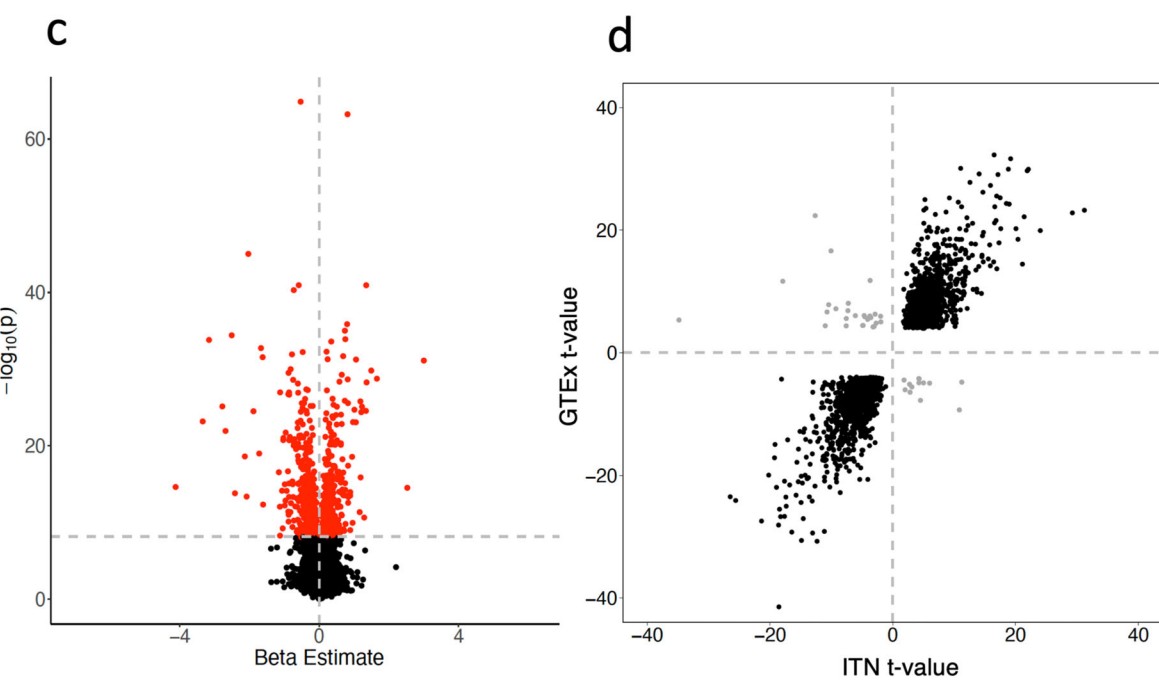

**Fig. 1 | Identifying eQTLs in patients with psoriasis. a** PAUSE clinical structure and sampling strategy for the individuals used for eQTL analysis. **b** Heatmap of lesional and non-lesional sample number per subject before and after treatment. **c** Volcano plot of eQTL effects for the significantly associated SNP for each gene (red color indicates $p < 6.7\text{e}-9$). **d** Concordance of ITN PAUSE deduplicated lead variant-gene pairs with significant eQTLs observed in the GTEx consortium[28] (FDR < 0.05) of 449 individuals. Each point represents a significant SNP–gene pair in this study. Concordant pairs are colored in black, discordant pairs are colored in gray.

Fig. 1d and Supplementary Data 3). Specifically, we observed concordant eQTL effects for several interesting eGenes with known associations with psoriasis risk. Consider the rs62121377 eQTL for *LGALS7* (beta = −1.60, *p* = 4.47e−13) and the rs2927608 eQTL for *ERAP2* (beta = 1.06, *p* = 5.60e−32) (Supplementary Fig. 3). Previous studies have shown that *LGALS7* expression is reduced in lesional skin from patients with psoriasis and that this reduction is associated with keratinocyte hyperplasia via increased cytokine (IL-6 and IL-8) expression and ERK signaling[41]. Similarly, studies have shown SNPs in *ERAP1* and *ERAP2* predict risk for development of psoriasis in an HLA dependent manner[42].

Using coloc[43], we performed colocalization analysis to assess the probability of sharing one common causal variant (i.e., PP.H4) between PAUSE skin eGenes and psoriasis GWAS variants[19]. We also calculated linkage disequilibrium (LD) $r^2$ between the lead eSNPs and psoriasis GWAS hits based on European population (EUR) in 1000 Genomes Project[44]. We noted 6 of our eGenes colocalize (PP.H4 > 0.75) with psoriasis loci, and 5 lead eSNPs have $r^2 > 0.5$ with psoriasis variants (Supplementary Data 4). For example, eSNPs influencing *IFNLR1* expression colocalize with psoriasis variants (PP.H4 = 0.834) and the lead eSNP rs59960858 is in almost perfect LD ($r^2 = 0.961$) with the psoriasis risk allele rs7552167 (Supplementary Data 4). *IFNLR1* encodes a subunit of a cytokine receptor and has been shown to exert antiviral effect in the context of psoriasis skin barrier breakdown[45]. The gene has also been mapped to psoriasis risk alleles across different studies[19,46–48]. Other colocalizing loci included eSNPs affecting expression of LCE3C, MTMR9, CTSW, SNX32, and RNA gene ENSG00000255389 (Supplementary Data 4).

To investigate the functional impact of psoriatic skin *cis*-eQTLs in inflammatory skin diseases more broadly, we expanded our analysis to include eczema and systemic scleroderma risk loci. We extracted GWAS summary statistics for the two additional traits and performed colocalization analysis of the identified *cis*-eQTLs and disease GWAS loci to identify potential shared causal variants[49,50]. Our results revealed 10 colocalizing loci, including 7 eSNPs colocalizing with eczema risk loci, and 3 colocalizing with scleroderma risk loci (Supplementary Data 4). The eGene that most strongly colocalized with eczema risk alleles was *PGLYRP4* (PP.H4 = 0.989) (Supplementary Data 4), which is an innate immunity-related gene encoding for peptidoglycan recognition protein. The deficiency of Pglyrp4 in mice has been reported to be involved in the development of eczema through reduced recruitment of Tregs and increased activation of Th17 responses[51]. In addition, eSNPs affecting *PDHB* expression colocalize with scleroderma risk alleles (PP.H4 = 0.992; Supplementary Data 4). *PDHB* encodes for a subunit of pyruvate dehydrogenase complex, which plays an essential role in metabolism and has been associated with cancer and neurological diseases[52,53]. Additionally, a study conducted in mice showed that pyruvate dehydrogenase deficiency (PDH) can lead to metabolic shift in keratinocytes which may in turn result in loss of epidermal stem cells[54]. The overlap between psoriatic *cis*-eQTLs and risk loci for other skin conditions provides another way of confirming and exploring the important inflammation-related signal present in our data.

**Assessing skin inflammation with transcriptional data.** We hypothesized that gene regulation was altered in active psoriasis skin lesions compared to non-lesional skin. In this case, eQTLs may have different magnitudes of effect as the inflammatory status of the skin changes. As psoriatic lesions represent active skin inflammation, we can compare lesional versus non-lesional skin biopsies to define a signature of skin inflammation. However, in the PAUSE trial, lesional and non-lesional status were determined on the first visit; future lesional or non-lesional biopsies were taken from the same site based on its appearance at the first visit. Therefore, the lesional status provided by the clinicians only indicates the presence of skin inflammation

at the first visit and does not necessarily reflect its status at subsequent visits, where local inflammation may have abated after treatment. In this trial, lesional status of the skin biopsies are subject to change, as participants showed clinical improvement in lesional skin following treatment with ustekinumab[11,12]. PCA of samples reveals that at baseline the lesional and non-lesional samples separate along PC1 and PC2; however, at subsequent timepoints following ustekinumab treatment, the baseline lesional samples were less distinguishable from baseline non-lesional samples along the same PCs (Supplementary Fig. 4a).

We therefore defined a skin psoriatic inflammation transcriptional score (SPITS) based on lesional and non-lesional biopsies at baseline, recognizing that biopsies undergoing active inflammation should have higher inflammation signatures that could be captured by gene expression level. We applied linear discriminant analysis (LDA), using the skin transcriptional data from the first visit (baseline) as training set (N = 140); these samples had their status determined by a clinician ("Methods"). We applied this classifier to post-first visit samples, which were unlabeled (N = 235) ("Methods"). We calculated a SPITS for each sample based on the first 48 RNA-seq PCs (>90% variance explained) (Supplementary Fig. 4b). Positive SPITS corresponded to lesional-like (i.e., inflamed) samples, while negative SPITS corresponded to non-lesional-like (i.e., less inflamed) samples—which presumably includes non-lesional and resolving lesional samples. Based on 10-fold cross-validation, we observed that SPITS was 95.00% (s.d. = 4.82%) accurate at separating the baseline lesional and non-lesional skin samples (Supplementary Fig. 4c).

Based on SPITS, we assigned the post-first visit samples to lesional-like SPITS-positive or non-lesional-like SPITS-negative statuses. Following treatment with ustekinumab, a total of 111 biopsies from resolving lesional skin areas demonstrated a negative SPITS score that resembled the baseline non-lesional biopsies, while 60 post-treatment biopsies from lesional areas demonstrated positive SPITS scores, most likely reflecting incomplete response to ustekinumab or clinical recurrence of psoriasis in the same location following ustekinumab withdrawal. In contrast, almost all non-lesional samples remained consistently negative at baseline, during treatment, and after withdrawal of study lead-in medication (Fig. 2a and Supplementary Fig. 5a).

To compare SPITS-based sample classification to non-psoriatic samples, we focused on the top 30% of genes most highly expressed in non-psoriatic GTEx skin samples (N = 2195 genes)[28]. We found that median expression of these genes was strongly correlated between baseline non-lesional samples and SPITS-negative lesional samples (R = 0.99). Expression in GTEx was more strongly correlated with non-lesional and SPITS-negative samples (GTEx and baseline non-lesional R = 0.78, GTEx and lesional SPITS negative R = 0.77) than with SPITS-positive samples (R = 0.67) (Supplementary Fig. 5b).

When comparing SPITS-positive to SPITS-negative samples, we found 457 upregulated and 384 downregulated genes (FDR < 0.05 and |log2FC| > 1.5, "Methods," Supplementary Fig. 6a, and Supplementary Data 5). Many differentially expressed genes (DEGs) are classical markers of keratinocyte hyper-proliferation or were previously described in psoriasis, suggesting that SPITS is capturing biologically relevant phenomena. For example, *VNN3* has higher expression in SPITS-positive samples and is up-regulated in psoriasis and induced by psoriatic-related pro-inflammatory Th1/Th17 cytokines (e.g., IL-17)[55]. The SPITS-positive up-regulated keratin genes, *KRT6A*, *KRT16*, and *KRT17*, are also known to be over-expressed in psoriasis, and are important regulators of innate immunity in the epidermis[56].

To further investigate the biological meaning of SPITS, we performed gene ontology (GO) analysis with the SPITS DEGs. We found that the upregulated genes in SPITS-positive samples are most highly enriched in GO terms related to keratinocyte-related psoriasis pathology such as keratinization (proportion of gene set = 0.36, FDR = 4.05e−17) and keratinocyte differentiation (proportion of gene

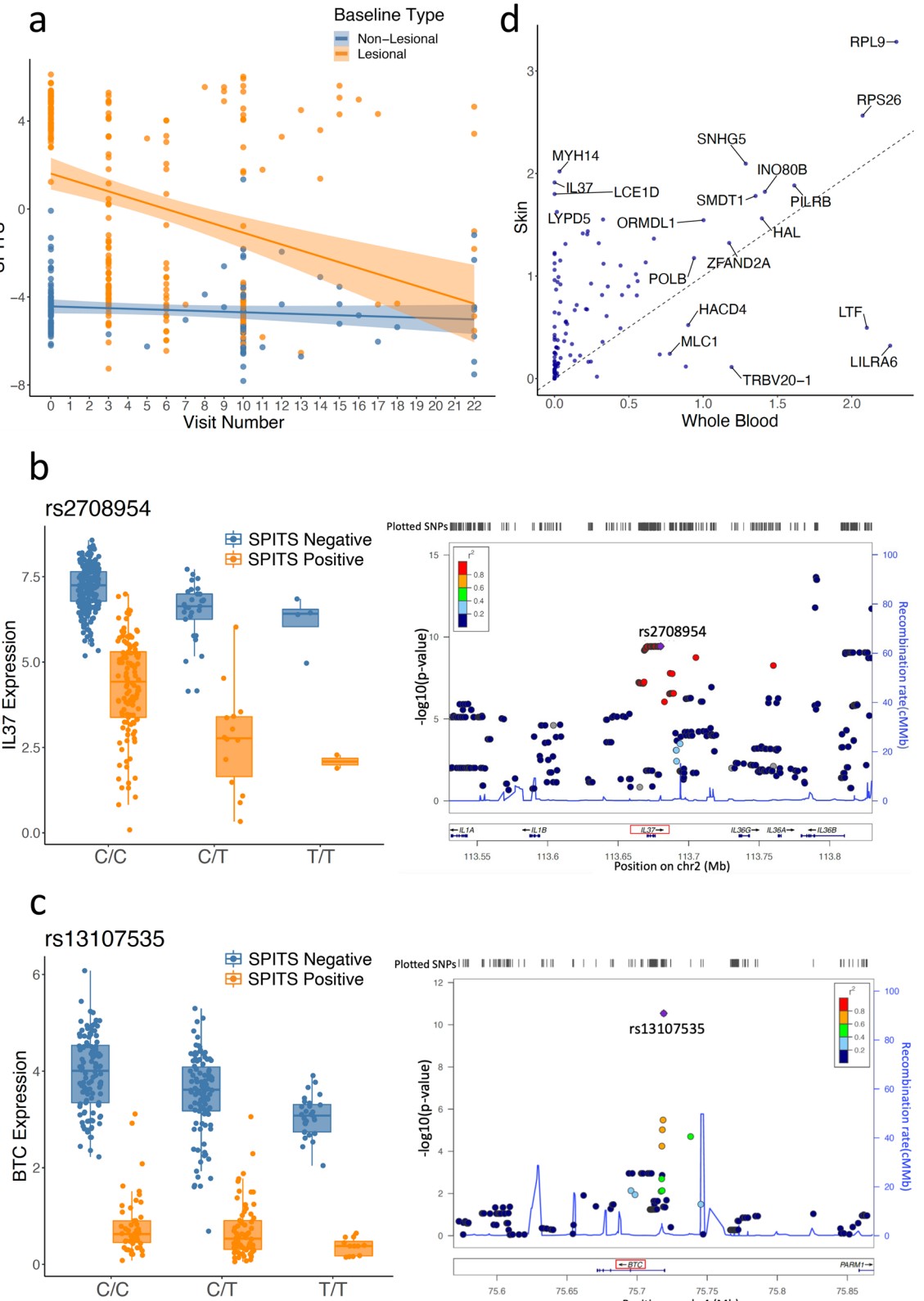

**Fig. 2 | eQTL interactions with inferred inflammation status (i.e., SPITS). a** At baseline, SPITS clearly separates lesional (SPITS positive) and non-lesional (SPITS negative) samples. With treatment, SPITS of lesional samples shift toward negative over time. Points correspond to biopsy samples. **b, c** SPITS interactions with *IL37*, *BTC* eQTL plotted with respect to rs2708954, rs13107535 genotype (left) and LocusZoom[95] regional association plots (right). The middle line in the box plots show medians, and the hinges correspond to the 25th and 75th percentiles. The whiskers extend the largest and smallest value no further than 1.5 × IQR from the hinges (*n* = 375 samples). **d** Median skin and whole blood expression of interacting eGenes from GTEx consortium. Points correspond to interaction eGenes.

set = 0.18, FDR = 3.39−14), or immune cell infiltration such as neutrophil chemotaxis (proportion of gene set = 0.19, FDR = 2.55e−11) and granulocyte chemotaxis (proportion of gene set = 0.17, FDR = 1.33e−11, Supplementary Fig. 6b). On the other hand, the genes upregulated in SPITS-negative samples are enriched in intermediate filament-related pathways, such as intermediate filament organization (proportion of gene set = 0.33, FDR = 8.07e−12, Supplementary Fig. 6b).

Given the role of IL-17 in inducing a psoriatic-like response in human keratinocytes (KC), we also calculated an IL-17 pathway score for each sample using a previously reported list of genes induced by IL-17 in human keratinocytes ("Methods")[57]. The keratinocyte IL-17 pathway score is strongly correlated with SPITS ($r = 0.92$, $p < 1e−26$, Supplementary Fig. 6c, d). We repeated the analysis with a curated IL-17 signaling pathway gene set that encompasses genes from the IL-17 cytokine family and their corresponding receptors and noted that the correlation between this systemic IL-17 pathway and SPITS is much weaker ($r = 0.20$, $p = 1.18e−04$, Supplementary Fig. 6c, d)[58]. These results demonstrate that SPITS reflects a local, but not systemic, response to IL-17 and other inflammatory factors in skin.

## Defining dynamic eQTLs whose effects are modulated by skin inflammation

Next, we identified dynamic eQTLs (i.e., eQTLs modulated by different contexts) with a statistical interaction model. For the 953 *cis*-eQTLs identified in our data set, we examined whether inflammation status (i.e., SPITS-positive vs SPITS-negative) alters the relationship between genomic variation and gene expression (i.e., SPITS status–eQTL interactions).

For the binary SPITS status interaction model, we compared its fit to one without the interaction term using likelihood ratio test. Both models included genotype and SPITS inflammation status as main effects (to capture baseline eQTL effects and differential expression, respectively), as well as the other covariates we used in our original eQTL analysis ("Methods"). To evaluate whether including PCs would be sufficient for correcting potential confounding factors such as age or sex, we re-ran the interaction analysis including age and sex as fixed effects. We found that including age and sex in addition to PCs has almost no effect on the interaction betas ($R = 1$, $p < 1e−15$, average change in nominal betas = 1.4%), suggesting the principal components are capturing these known confounders (Supplementary Fig. 7a). We also considered the possibility that the inclusion of expression PCs might be reducing the power to detect interactions. To further explore this, we repeated the SPITS status interaction analysis without correcting for principal components 1 and 2, which are significantly correlated with SPITS status (PC1-SPITS status $R = 0.64$, $p = 1.94e−45$; PC2-SPITS status $R = 0.66$, $p = 2.17e−47$). We found the betas for the interaction term are highly correlated ($R = 0.99$, $p < 1e−15$, Supplementary Fig. 7b), and that including these PCs increased the number of eQTLs discovered (90 at FDR < 0.05 without PC1 and 2, versus 98 at FDR < 0.05 with PC1 and 2).

We observed a total of 98 interactions at FDR < 0.05 when comparing SPITS positive to SPITS negative samples. We observed 116 interactions at a more permissive FDR < 0.2 threshold (Supplementary Data 6), Permuting SPITS status, we found the results to be robust at both FDR < 0.2 and FDR < 0.05 (all permutation $p < 0.01$ from 100 permutations). The eQTL interaction with local inflammation defined by SPITS were all significant whether we conditioned on treatment arm or not (Supplementary Data 6), suggesting that local inflammation rather than systemic treatment was driving interaction effects.

We considered whether a less stringent p-value threshold for original cis-eQTLs may have enabled us to identify more eQTL interactions. Arguably, a Bonferroni corrected $p < 6.69e−9$ threshold is too stringent since it does not account for LD within loci which would reduce the effective number of tests being conducted. To explore the effect of a more relaxed threshold, we ran a separate analysis

using a lowered threshold ($p < 1e−7$). Further, we explored skin inflammation–eQTL interactions using a similarly relaxed threshold. Lowering the threshold by more than 100-fold led to the discovery of 1266 significant eQTLs ($N = 953$ if Bonferroni p-value is applied), 145 SPITS interactions at FDR < 0.20 ($N = 116$ if Bonferroni p-value is applied) and 115 SPITS interactions at FDR < 0.05 ($N = 98$ if Bonferroni p-value is applied) (Supplementary Fig. 8a). This relaxation resulted in only a moderate increase in signal with small effect sizes and larger FDRs (Supplementary Fig. 8b). We therefore used the more conservative Bonferroni threshold to ensure that we were confident in the eQTL main effect before testing that effect for an interaction. We also tested a continuous SPITS score, which detected fewer eQTL interactions (88 interactions at FDR < 0.05, 93 interactions at FDR < 0.20). An alternative strategy testing interactions with lesional versus non-lesional labels as assigned at baseline resulted in even fewer eQTLs (39 eQTL interactions at FDR < 0.5, 42 eQTL interactions at FDR < 0.2). These results suggested that local inflammation at the time of the biopsy was more informative than the status of the skin during baseline assessment. We thus chose to use the binary SPITS status-eQTL interactions for subsequent analyses. We further subdivided SPITS interactions into magnifiers ($n = 51$, FDR < 0.2), where skin inflammation (SPITS-positive) increases the size of the eQTL effect, and dampeners ($n = 65$, FDR < 0.2) where inflammation decreases the size of the eQTL effect.

We also detected 98 significant IL-17 keratinocyte pathway score interactions at FDR < 0.2 (86 at FDR < 0.05), among which 97 overlapped with SPITS interactions (Supplementary Data 6). In contrast, the same analysis approach revealed no significant eQTL interactions with treatment arm and found few interactions with PASI score at FDR < 0.05 ($N = 2$) or FDR < 0.20 ($N = 5$) (Supplementary Data 6). The number of PASI interactions remain the same after adjusting for SPITS status, possibly suggesting that dynamic gene regulation is more strongly connected to local skin inflammation, rather than global disease burden, as captured by PASI.

As an example of a SPITS-eQTL interaction, IL37 expression is associated with rs2708954 (main effect beta = −0.77, $p = 3.73e−10$) and this negative effect is significantly magnified in inflamed skin samples (SPITS interaction beta = −0.79, FDR = 7.73e−7, Fig. 2b). The beta in SPITS positive samples is the sum of the main effect and interaction effect from the model with interaction term (beta = −0.77 − 0.79 = −1.56). *IL37* is an anti-inflammatory cytokine, and has been shown to mitigate the inflammation in psoriasis experimental models by suppressing proinflammatory cytokines[59,60]. Another example is *BTC*, which belongs to the epidermal growth factor (EGF) family. *BTC* expression is associated with rs13107535 (main effect beta = −0.41, $p = 2.86e−11$), and the effect is dampened in inflamed skin samples (interaction beta = 0.49, FDR = 2.46e−10, Fig. 2c). *BTC* is reported to be downregulated in psoriatic lesions and has an important role in skin morphogenesis and homeostasis[61–64]. Our SPITS-eQTL interactions captured disease relevant signal whose upstream regulation may play an important role in the development or progression of psoriasis.

## Understanding the role of cell type in dynamic eQTLs

We sought to understand whether these dynamic eQTLs were related to specific cell types or cell states. Psoriatic lesions involve the infiltration and activation of haematopoietically derived immune cells interacting with skin cells in the dermal and epidermal layers. We initially hypothesized that, due to higher levels of immune cell infiltration and activation, the dynamic eQTLs might be related to changes in immune cell state or activation rather than dermal and epidermal cells.

To assess this hypothesis, we obtained median gene-level TPM by tissue datasets from the GTEx consortium, and compared the median expression of interacting eGenes in skin with that in whole blood[28].

Among the 112 (out of 116 total eGenes with FDR < 0.2) interacting eGenes present in GTEx data, the vast majority have a higher expression level in skin ($n = 99$, 90.83%) than in whole blood ($n = 10$, 9.18%) ($p < 0.001$) (Fig. 2d). These results suggested that eQTL interactions may be predominantly related to changes in dermal and epidermal cell states, possibly as a result of interactions with immune cells or cytokines. We recognized however that GTEx skin and blood data capture unperturbed skin cell states and CD45+ immune cells in a resting state. Hence, it is critical to investigate this question in cells obtained from skin tissue itself.

We examined expression of eQTL interaction genes using a dataset of single cells obtained from skin, including psoriatic skin lesions[32]. With this psoriatic single-cell data, we quantified the expression levels of interacting eGenes in each cell type. Among the eGenes that were present in the single-cell data ($N = 84$), we found that some eGenes are expressed across cell types (e.g., *RPL9*, *RPS26*), while some are more cell-type-specific (e.g., macrophage-specific *SIGLEC12*) (Supplementary Fig. 9). Consistent with our findings in GTEx data, we observed that the majority of the eGenes are maximally expressed in dermal and epidermal cells ($N = 62$) including keratinocytes ($N = 22$), melanocytes ($N = 14$) and fibroblasts ($N = 11$), while only 22 of 84 genes were immune cell-specific (Fig. 3c).

## Cell fraction deconvolution and eGene expression across cell types

Given that our inflammation-interacting eGenes are mostly expressed in dermal and epidermal cells, we hypothesized that some of our observed interactions could reflect variability in cell type or cell state proportions altered by inflammation status. To estimate these proportions from bulk RNA-seq, we used CIBERSORTx[65] to deconvolute the relative proportions of 14 cell populations (7 immune, 7 dermal and epidermal) ("Methods"). For this analysis, we used the previously mentioned psoriatic single-cell dataset. In brief, for each cell type, we sampled at most 1000 cells from the single-cell data[32], trained the model on these expression profiles, and inferred cell fractions in each PAUSE trial bulk RNA-seq sample with a linear regression model[65]. We observed 10/14 cell types are differentially abundant between SPITS-positive and SPITS-negative samples by Wilcoxon rank sum test ($p < 0.001$ for 9/10, $p < 0.05$ for 10/10) (Fig. 3a). For example, SPITS-positive samples contain higher proportions of keratinocytes (mean percent proportion 38.71% in SPITS-positive vs 17.57% in SPITS-negative) and macrophages (3.05% in SPITS-positive vs 0.02% in SPITS-negative), while SPITS negative samples have more fibroblasts (12.24% in SPITS-positive vs 23.22% in SPITS-negative), pericytes (6.09% in SPITS-positive vs 10.52% in SPITS-negative) and Langerhans cells (4.07% in SPITS-positive vs 6.18% in SPITS-negative) (Fig. 3b). By contrast, estimated T cell proportions are not differentially abundant (11.13% in SPITS-positive vs 10.93% in SPITS-negative) between the SPITS positive and SPITS negative samples. These differences might be related to the use of immunosuppressive agents. Additionally, the bulk RNAseq data comes from a punch biopsy which cannot provide spatial context of T cells residing in epidermis vs. dermis, nor can it provide the functional status of T cells, which could be very different in SPITS-positive vs. SPITS-negative samples.

Given the crucial roles of macrophages in skin inflammation and wound healing[3], we further explored the trajectory of macrophage proportions before and after treatment. To do that, we split the samples from first visit (before treatment) and after first visit (after treatment) and compared the inferred relative abundance of macrophages in SPITS positive negative and positive groups, respectively. We observed that after treatment, there is a significant decrease in macrophage abundance in both SPITS groups ($p$ value = 6.39e−3 in SPITS negative samples, $p$ value < 1e−10 in SPITS positive samples), while the change is more prominent in SPITS positive samples (mean fraction drops by 3.36e−4 in SPITS

negative samples, by 1.15e−2 in SPITS positive samples, Supplementary Fig. 10).

Although most major cell type proportions are altered by inflammatory states, some cell types are more important in defining inflammation status. To better understand the relative importance of the cell types present in the bulk RNAseq, we fit logistic regression models with SPITS status as the outcome and cell type proportions as predictors ("Methods"). Using forward stepwise selection, we identified cell types associated with SPITS based on a decrease in deviance (entry significance level = 0.05, "Methods"). Among all the cell types included in the model with lowest deviance, the proportions of keratinocytes lead to the highest decrease in deviance (76.89%) (Supplementary Fig. 11a, b). Since the proportion of macrophages and fibroblasts rank second and third respectively with regards to contribution to univariate logistic regression model, they are also likely informative in distinguishing the samples (Supplementary Fig. 11c). With LDA, the proportions of KC, along with fibroblast and macrophage were excellent predictors in differentiating SPITS-positive and SPITS-negative samples (area under the receiver-operator curve [AUC] = 0.9828, Supplementary Fig. 11d).

As expected, the predicted cell type fractions suggested a shift in cell populations between SPITS-positive and SPITS-negative samples. We also found that the interacting eGenes were mostly expressed in the same differentially abundant cells (e.g., keratinocyte, fibroblast, melanocyte) between SPITS-positive and negative-samples (Fig. 3c). We next assessed whether the eQTL interactions were also linked to these cell type proportions, which might be the result of a higher level of inflammation. We performed interaction analysis with three cell types of largest predicted proportions: keratinocytes (mean percent proportion = 25.07%), fibroblasts (19.33%) and T cells (11.00%). We identified cell type–eQTL interactions and compared them with the SPITS inflammation status interactions. In total, we found 103 keratinocyte proportion-eQTL interactions, 65 fibroblast proportion-eQTL interactions and 11 T cell proportion–eQTL interactions at FDR < 0.20. Among them, 94 keratinocyte interactions (91.26%), 58 fibroblast interactions (89.23%) and 4 T cell interactions (36.36%) overlap with the 116 SPITS inflammation status interactions (Supplementary Fig. 12). This supports our initial hypothesis that interactions correlate with variability in cell type proportions when SPITS status changes, and again indicates the importance of the dermal and epidermal compartments in psoriasis.

Given that macrophages play a crucial role in managing skin inflammation and promoting wound healing[3], and that we reported higher proportion of macrophages in SPITS-positive samples compared to SPITS-negative, we further assessed interactions with macrophage proportion. We found 71 interactions at FDR < 0.05, and 82 interactions at FDR < 0.20. Among them, 69/71 (97.2%) and 77/82 (93.90%) were overlapping with SPITS interactions, respectively. Among the macrophage proportion-interacting eGenes, only one of them, *SIGLEC12*, is highly expressed in macrophages (logFC = 4.76, Fig. 3c). Many of these other eGenes may be occurring in non-macrophage cell-types; thus, eQTL interactions with macrophage proportion may be due to the proportion being a marker of inflammation level.

## Detecting Molecular Pathways underlying eQTL interactions

To unravel the molecular pathways underlying the SPITS interactions, we first identified the functional pathways involving the eGenes. To do that, we ranked the eGenes by the absolute value of interaction coefficient, and performed GSEA using fgsea and MSigDB gene sets[58,66–68]. Although none of the gene sets tested passed FDR < 0.05 from 10,000 permutations, it is interesting to note that the eGenes overlap with skin-related pathways like "formation of the cornified envelope" (permutation $p = 6.90e−03$, normalized enrichment score = 1.73, Supplementary Fig. 13) and "keratinization" (permutation $p = 7.70e−03$,

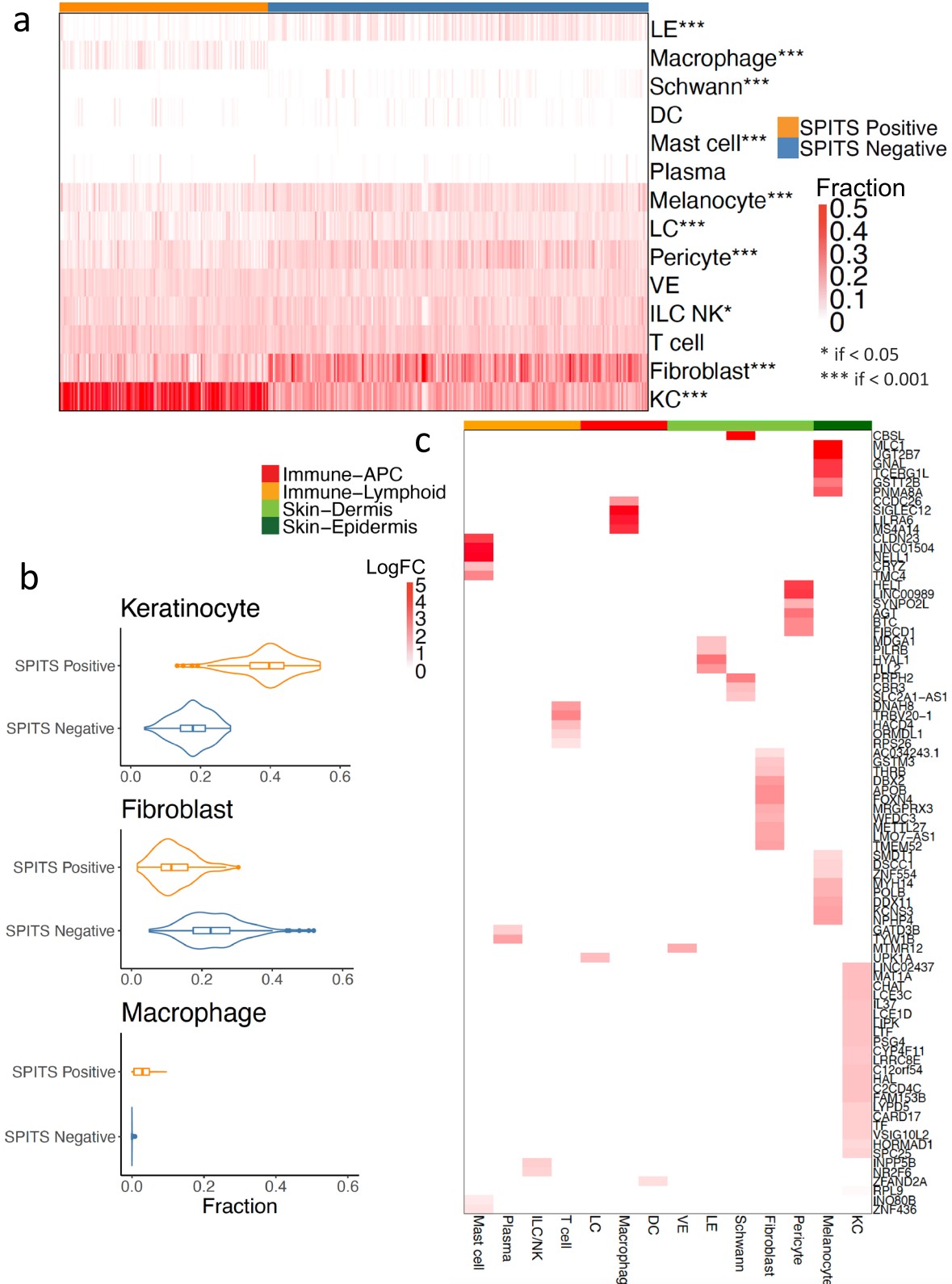

**Fig. 3 | Cell fraction deconvolution and eGene expression across cell types.**
**a** CIBERSORTx[65] predicted cell proportions in the samples. **b** Fractions of KC, fibroblast and macrophage in SPITS positive and SPITS negative samples. The middle line in the box plots show medians, and the hinges correspond to the 25th and 75th percentiles. The whiskers extend the largest and smallest value no further than $1.5 \times IQR$ from the hinges ($n = 375$ samples). **c** The 84 eGenes with the highest relative expression as identified by the log fold change of the expression against the per-gene mean expression across the 14 cell types. LE lymphatic endothelium, ILC_NK innate lymphoid cells and natural killer cells, DC dendritic cells, LC Langerhans cells, VE vascular endothelium, KC keratinocyte, APC antigen presenting cells.

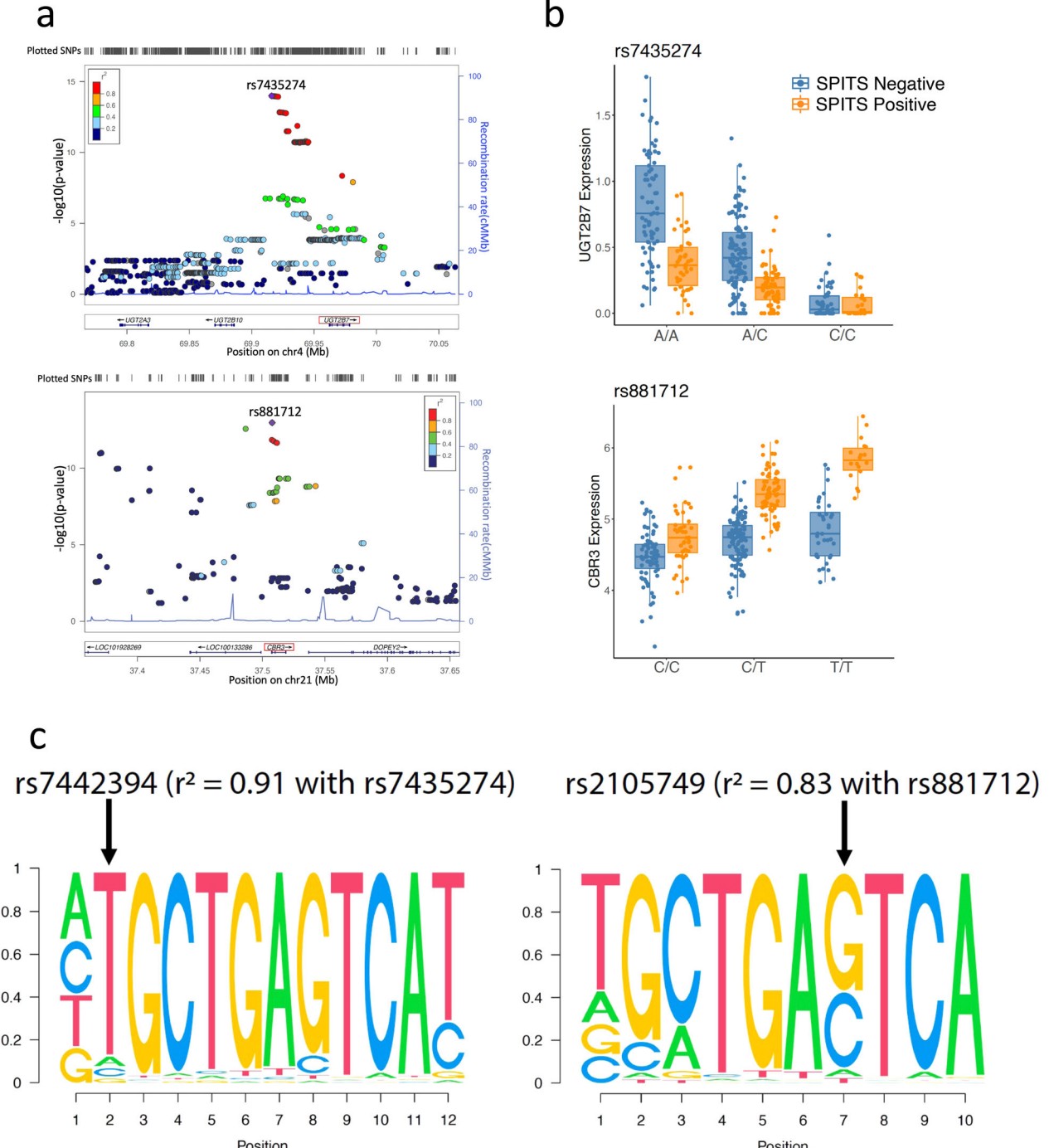

**Fig. 4 | Regional association plots and motif enrichment analysis. a** Regional eQTL association plots of rs7435274-*UGT2B7*(top) and rs881712-CBR3(bottom), two eQTL interactions with eGenes overlapping with NRF2 gene set. **b** SPITS interaction with the *UGT2B7*(top), *CBR3* (bottom) eQTLs plotted with respect to rs7435274, rs881712 genotypes, respectively. The middle line in the box plots show medians, and the hinges correspond to the 25th and 75th percentiles. The whiskers extend the largest and smallest value no further than 1.5 × IQR from the hinges ($n = 375$ samples). **c** The ARE motifs disrupted by interacting eQTLs. Arrows indicate positions of the motif disrupted by eSNPs in strong LD with the interaction eSNPs in *UGT2B7* (left), *CBR3* (right) eQTLs.

normalized enrichment score = 1.72, Supplementary Fig. 13). We also found that the NRF2 pathway (permutation $p = 1.16e{-}02$, enrichment score = 1.71, Supplementary Fig. 13) overlaps with the interacting eGenes *GSTM3, CBR3, UGT2B7* and *GSTT2* (Fig. 4a, b). NRF2 is a master transcription factor regulating expression of antioxidative genes. It is activated under oxidative stress and also plays a key role in repressing inflammation[69,70]. Previous studies report that upon stimulation with IL-17 in psoriasis mouse models, Nrf2 promoted keratinocyte proliferation by up-regulating keratin genes[71].

We went on to identify potential common regulatory elements overlapping interacting eQTLs. We used HOMER[72] to assess overlap between transcription factor binding motifs and the eQTL interaction SNPs. For the input sequences, we included the eSNPs (FDR < 0.20) and the SNPs in high linkage disequilibrium (LD, $r^2 > 0.8$) within 500kB window around the TSS using the EUR reference panel, and then defined +/− 20 bp intervals around these SNPs[44]. We merged the sequences into a non-overlapping set of intervals before running motif enrichment analysis. We compared the target sequences to

background sequences generated from SNPs matched to interaction eSNPs by MAF, number of SNPs in LD, and gene density[73]. We observed enrichment of ARE (antioxidant response element) motifs that NRF2 binds in response to oxidative stress (Fig. 4c, left motif enrichment: $p = 1e−7$, right motif enrichment: $p = 1e−2$)[74,75]. Moreover, the SNPs rs7442394 and rs2105749, which are both in high linkage disequilibrium ($r^2 = 0.91$ and $r^2 = 0.83$, respectively) in EUR population with the interaction eSNPs for *UGT2B7* and *CBR3*, disrupt the ARE motifs[44] (Fig. 4c). Alongside the (non-significant) enrichment of interacting eGenes in the NRF2 pathway, this finding indicates a potential role of NRF2 in mediating interactions with inflammation state in psoriasis.

### *LCE3C* eQTL interaction colocalizes with psoriatic risk locus

As mentioned previously, we detected 6 colocalizations between our eQTL variants and psoriatic variants. Among them, we reported a posterior probability of 0.996 that skin eQTLs and psoriasis risk alleles share a same casual variant at the *LCE3C* locus. Furthermore, the eSNP rs1491377616 influencing *LCE3C* is in perfect LD ($r^2 = 1$) with the psoriasis-associated variant rs6677595 (Supplementary Data 4). Indeed, the rs1491377616-*LCE3C* eQTL (main effect beta = −2.69, $p = 1.14e−22$) has the most significant interaction with SPITS score, and its effect is dramatically magnified (more than doubled) in skin samples with more inflammation (interaction beta = −3.86, FDR = 1.59e−79, Fig. 5a, b). *LCE3C* expression is restricted to epidermis (Fig. 3c), and is induced in both psoriatic and stimulated skin barrier disruption[76]. Moreover, the *LCE3C* deletion allele is a widely replicated psoriasis risk factor, possibly related to inappropriate skin repair as a result of the deletion[76–78]. Consistent with previous findings, we found increased levels of *LCE3C* in inflammatory skin biopsies (SPITS-positive), and the eQTL interaction analysis allows us to demonstrate potential differential regulation of *LCE3C* in epidermal cells, during inflammation, extending the mechanistic relationship between *LCE3C* and psoriasis[76].

## Discussion

Randomized clinical trials, such as the PAUSE trial, provide a unique opportunity to query the molecular basis of a disease. Genomic data from clinical trials offer a well-controlled environment with high internal and external validity. Specifically, the PAUSE study provided the opportunity to explore *cis*-eQTLs, genetic variants that alter the expression level of a nearby gene, across matched psoriatic lesional and non-lesional skin. These interactions between eQTL and inflammation status can illuminate gene regulatory mechanisms underlying the switch from normal appearing skin to psoriatic lesional skin, or alternatively reflect changes in cell states. With data from a controlled clinical trial, we were able to separate out systemic effects, such as medications, from local inflammatory effects.

This study is the first of its kind to examine the effect of SNPs on expression in psoriatic skin in a clinical setting. While the role of immune activation on eQTLs in immune cells is well-established, its effect on tissue cell types is much less well understood. Our study highlights the dynamic nature of gene regulation within the dermal and epidermal compartments in disease. We found the eGenes whose effects are modulated by disease-mediated inflammation (i.e., SPITS) are more highly expressed in skin cells such as keratinocytes, fibroblasts, and melanocytes, compared to immune cells. Furthermore, deconvolution analysis revealed that differences in epidermal and dermal cell proportions, not immune cells, was associated with differences between inflammatory and non-inflammatory samples. In the chronic disease process of psoriasis, cytokines produced by T cells interact with epidermal and dermal cells, which in response become T cell co-activators, sustaining adaptive immunity[3]. Previous transcriptomic studies of psoriasis reported that, in the mix of genes that are differentially expressed in psoriasis lesions, most are specific to

keratinocyte (56% of upregulated DEG), epidermis (14% of downregulated DEGs) and dermis (4% of downregulated DEGs), while fewer are found in immune cells (14% and 11% of upregulated DEGs are explained by infiltration of T cells and macrophages, respectively)[79]. Our results consistently point to more changes in gene regulation in skin cell types compared to immune cell types in the context of inflammation.

The role of immune cells, such as IL-17-producing T cells, is well-established in the pathogenesis of psoriasis, and indeed some therapeutic interventions target IL-17. In our study, a minority of identified eQTL interactions could be attributed to immune cell populations. However, by defining the keratinocyte IL-17 response pathway score of each sample, we found 98 IL-17 pathway–eQTL interactions, in which 97 overlapped with SPITS–eQTL interactions. IL-17 is the major effector cytokine in the pathogenesis of psoriasis, and its primary targets include keratinocytes[80]. Our results again show that IL-17 production may lead to an inflamed environment that causes regulatory changes in epidermal and dermal cells.

We also observed few eQTLs modulated by the therapeutic agents used in the treatment arms of the trial or global disease activity, whether or not we controlled for local skin inflammation (i.e., SPITS), suggesting that altered regulatory effects were largely driven by local factors, rather than systemic ones. This ability to separate local and systemic factors was only possible because of the clinical trial setting in which this study was conducted.

Our results point to NRF2 as a potential mediator of many regulatory interactions with inflammation. Studies have reported the important role of NRF2 both in immune responses and keratinocyte proliferation. It is well established that NRF2 is activated under stress conditions, and through different pathways, is able to exert cytoprotective effects and repress inflammation[69]. Conversely, recent studies have also implicated NRF2 in the activation of the pentose phosphate pathway and the proliferation of keratinocytes that might contribute to psoriasis pathogenesis[81]. In mouse psoriasis models, Nrf2 was reported to translocate to the nucleus in response to inflammatory cytokines such as IL-17 and IL-22, and upregulate the expression of key keratin genes in lesional epidermis, ultimately leading to keratinocyte proliferation[71]. The role of NRF2 pathway in psoriasis, whether it induces a cytoprotective mechanism or promotes keratinocyte hyperproliferation remains to be seen. Future studies to examine the NRF2 pathway in psoriasis pathogenesis may point to therapeutic avenues for the disease.

We detected *LCE3C* as the most significant interaction eGene with local skin inflammation (interaction beta = −3.86, FDR = 1.59e−79). Studies have reported the association between deletion of *LCE3C* and psoriasis susceptibility across different populations[76,78,82]. The gene is proposed to play a role in skin barrier repair, possibly through its antibacterial activity, and has increased expression in psoriatic lesions and induced skin injury[76,77,83]. Our study consistently reported higher *LCE3C* levels in inflamed skin biopsies and found that the rs1491377616-*LCE3C* eQTL effect is magnified in inflamed skin, suggesting differential regulation of the gene in psoriatic lesions. In addition, eSNPS influencing LCE3C expression colocalize with psoriasis variants, and the lead eSNP rs1491377616 is in perfect LD with rs6677595, a psoriasis risk allele. Our results not only point to a possible role for *LCE3C* in the genetic risk of psoriasis, but also suggest local skin inflammation could modify this pathway to contribute to psoriatic disease.

We also performed colocalization analysis between psoriatic skin eQTLs and GWAS of other inflammatory skin diseases, like eczema and scleroderma. This comparative approach allowed us to identify potential shared causal variants between psoriatic skin eQTLs and other skin diseases. For instance, we found that psoriatic skin eSNPs and eczema risk loci colocalize at *PGLYRP4*, which has been linked to eczema development in mouse models through Th17 activation[51]. Similarly, we observed colocalization at the *PDHB* gene with

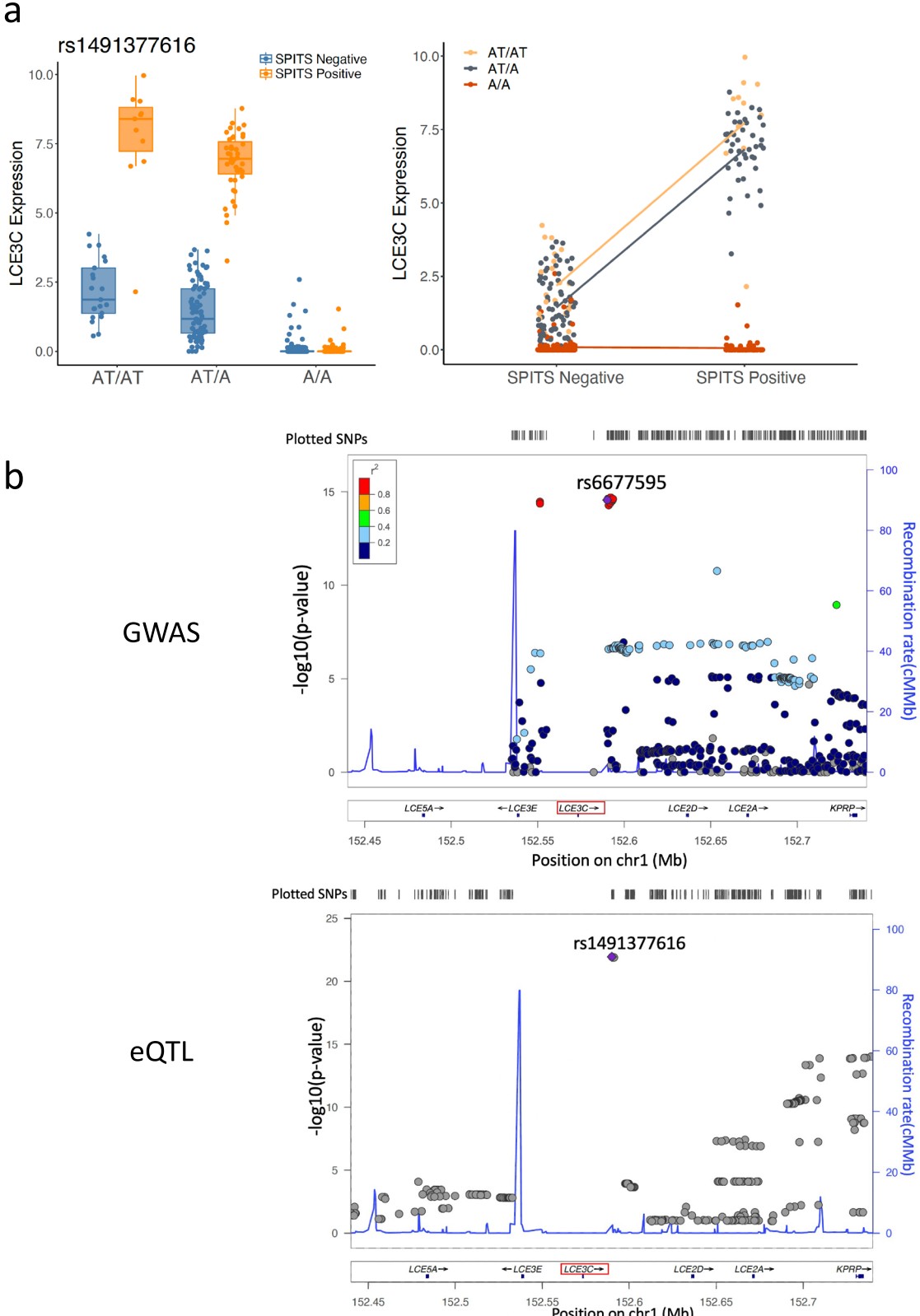

**Fig. 5 | LCE3C interaction and regional association plots. a** SPITS-interactions with *LCE3C* eQTL plotted with respect to rs1491377616 genotype (left) and SPITS status of the sample (right). The middle line in the box plots show medians, and the hinges correspond to the 25th and 75th percentiles. The whiskers extend the largest and smallest value no further than 1.5 × IQR from the hinges (*n* = 375 samples). **b** Regional association plots with respect to *LCE3C* locus, with P-value obtained from previous psoriasis GWAS study[19] (top), and eQTL association (bottom). rs881712 and rs1491377616 are in perfect LD.

scleroderma. The gene encodes for a subunit of pyruvate dehydrogenase, and the PDH enzyme was reported to be associated with skin homeostasis in mice[52]. By examining colocalization of eQTLs across different skin diseases, we may gain a more comprehensive understanding of the genetic basis of these conditions and potentially identify common pathways that underlie these conditions or that can be targeted for therapeutic interventions. Our skin eQTL dataset thus provides a valuable resource for identifying potentially causal variants that are involved in the pathogenesis of inflammatory skin diseases.

One limitation to this study is that it is smaller than some of the large-scale eQTL studies[28,84], but is the first examining diseased skin tissues obtained from a clinical trial. The relatively small size might limit power to detect cis-eQTL main effects and interaction effects, and the fact that the cohort is largely European ancestry limits our ability to examine population-specific eQTL signals (Supplementary Fig. 14). Our power was bolstered, though, by having repeat samples, which reduces noise and enables detection of interactions[36]. Moreover, we employed a strict Bonferroni p-value threshold to suppress Type I error, but we may have also excluded many true positive signals that we would have been able to detect with decreased stringency or increased sample size. We acknowledge that our approach for eQTL discovery using a stringent Bonferroni corrected p-value threshold may be conservative and might reduce our ability to detect eQTLs with a modest effect. Consequently, this may reduce the number of interactions we observe. However, given the challenge of identifying interactions, we wanted to ensure that we were confident in the eQTL effect before testing that effect for an interaction. Additionally, all participants in the PAUSE study received immunomodulatory anti-IL-12, anti-IL-23 (ustekinumab), and the experimental group also received anti-CTLA-4 (abatacept) treatment during the trial. This context may have reduced our power to detect regulatory differences in immune cells between inflammatory and non-inflammatory samples, making it easier to detect the dermal and epidermal effects we highlight.

Psoriasis is an immune-mediated disease, driven by T cells and dendritic cells along with their associated cytokines and chemokines as the key players in activating keratinocyte proliferation[3,4,7]. Studies to understand the genetic basis of immune-mediated diseases have thus often focused on blood samples and assayed immune cells like T cells in the blood. However, these studies may be missing important gene regulatory effects in the tissue. Our study is the first of its kind focusing on eQTLs in lesional and non-lesional skin samples and may serve as motivation for other studies in the future to include inflammatory disease tissue samples.

## Methods

### Study design and ethical approval

The purpose of this study was to identify eQTLs in a cohort of individuals with psoriasis participating in the ITN PAUSE clinical trial (NCT01999868)[37]. The trial was described in detail by Harris et al. in the previous clinical paper[37]. Briefly, 108 participants with moderate to severe plaque psoriasis received ustekinumab at weeks 0 and 4 during the lead-in period. At 12 weeks, 91 patients who met the psoriasis response eligibility criteria (≥75% improvement in Psoriasis Activity and Severity Index [PASI]) were randomized 1:1 to blinded treatment with ustekinumab or abatacept. 45 participants received weekly subcutaneous abatacept from week 12 to week 39, while 46 participants continued to receive ustekinumab at week 16 and 28. Each group received the corresponding placebo to preserve blinding. Participants were followed for up to 88 weeks or until relapse (defined as a 50% loss of the PASI improvement achieved at week 12). A summary of participants included in this study is described in the published paper and Supplementary Table 1[37]. All participants provided written informed consent. The trial was conducted in compliance with the Declaration of Helsinki and was approved by the institutional review boards at all of the investigational sites (US: Dermatology Research Associates, Los Angeles, California; Northwestern University,Chicago, Illinois; Tulane University School of Medicine, New Orleans,Louisiana; University of Michigan, Ann Arbor; The Rockefeller University, New York, New York; Wake Forest University, Winston-Salem, North Carolina; Case Western University, Cleveland, Ohio; and University of Utah, Salt Lake City; Canada: Kirk Barber Research, Calgary, Alberta, and Innovaderm Research Inc, Montreal, Quebec).

### RNA-sequencing

We obtained skin punch biopsies from a representative active psoriasis lesion and a non-lesional area, placed in RNAlater, and stored frozen at −70 to −80 °C. We collected longitudinal biopsies repeatedly from the same area of skin. We isolated total RNA using Qiagen kits. We assayed samples with RNA Integrity Number (RIN) > 7 using RNA sequencing (Illumina HiSeq4000, paired-end 100 bp × 2 cycle, polyA selection of total stranded RNA) at a sequencing depth of ~47 million reads per sample. We aligned raw sequencing reads to the Ensembl GRCh38 v100 reference genome and quantified using kallisto v0.46.2. We summarized transcripts per million (TPM) counts using the tximport package to obtain gene-level TPM. We included genes with expression >0.1 TPM and >6 counts across at least 20% of samples in RNA-seq and eQTL analyses (27,100 genes). We excluded RNA-seq samples with RIN score <7.

### Genotyping

We genotyped 101 participants across 1,748,250 variants using the Infinium Multi-Ethnic Global BeadChip from Illumina. We removed 5 samples due to high missingness (>10%) and filtered SNPs with call rate <0.99, MAF < 0.05, or Hardy–Weinberg Equilibrium (HWE) $p < 1e-6$. We removed an additional 19 samples due to a lack of or low-quality RNA-seq data from these individuals, leaving 77 patients for eQTL analyses. Phasing and imputation were conducted using SHAPEIT[85] and minimac3[86]. We included reference data to phase samples from 1000 Genomes due to the small sample size of the study. After imputation, we removed SNPs with imputation quality ($R^2$) < 0.99 and MAF < 0.05, leaving 2,074,125 SNPs for further analysis.

### Colocalization analysis

For each of the significant cis-eQTL genes, we performed colocalization analysis with publicly available GWAS studies using coloc (v.5.1)[43]. We downloaded GWAS summary statistics of three diseases involving skin inflammation from GWAS Catalog (https://www.ebi.ac.uk/gwas/), including psoriasis (GCST005527), scleroderma (GCST009131) and eczema (GCST90044763). To detect colocalizing signals between eQTLs and GWAS variants, we matched the GWAS SNPs based on their effect/risk alleles to both reference and altered alleles of eQTLs, and used the posterior probability of sharing one common causal variant (i.e., PP.H4) > 0.75. We noted that for all three diseases we had p-values for each SNP. For scleroderma and eczema GWAS studies the effect coefficients were not publicly available. So for the coloc analysis, we calculated the probability of colocalization using coefficients and variance for psoriasis, and p-values for eczema and scleroderma. To assess linkage disequilibrium (LD) between the previously reported psoriasis-associated variants and our eQTL lead SNPs, we calculated LD r2 between them, and identified the eSNPs that have high LD ($r^2 > 0.5$) with psoriasis risk SNPs[19].

### Principal components analysis

The 2942 genes with mean expression and standard deviation >70% of all genes (i.e., top 30% most variable genes) were included to conduct principal component (PC) analysis using the prcomp function from the

stats R package[87]. We used the top 20 expression PCs for downstream analysis.

## Skin Psoriatic Inflammation Transcriptional Score (SPITS)

SPITS is a continuous score capturing the range of inflammation observed across lesional and non-lesional samples. It is defined by centered linear discriminant score from linear discriminant analysis (LDA); samples with positive SPITS values (i.e., positive linear discriminant score, above LDA line) have the highest level of inflammation (characteristic of lesional samples at baseline), and the samples with negative values (i.e., negative linear discriminant score, below LDA line) have low inflammation (characteristic of non-lesional samples at baseline). We denoted samples with positive SPITS to be SPITS positive, and samples with negative SPITS to be SPITS negative. In order to fit the LDA model, we used the **lda()** function from the MASS package[88]. The data were split into training set (first visit samples $N = 140$) and unlabeled validation set (post-first visit samples $N = 235$). The first 48 RNAseq PCs (accounting for >90% variance of the data) were used as the predictor variables and the biopsy type (lesional/non-lesional) as the response variable. The centered linear discriminant score of each sample was defined as SPITS.

To evaluate the predictive power of keratinocyte, macrophage and fibroblast proportions, we tested these cell types as predictors in LDA. 200 samples were randomly selected for the training set, and the rest were included in the test set. To evaluate the performance of the classifier, we used the roc() function from the pROC package[89].

## Differential gene expression analysis

We used the R package edgeR[90] to find the differentially expressed genes between SPITS-positive and -negative samples ($N = 375$). Genes with CPM > 1 in more than 20% of the samples were included for analysis. Differential expression is defined by FDR < 0.05 and |log2FC| > 1.5.

## Gene Ontology (GO) analysis

We performed pathway analysis of the SPITS differentially expressed genes (DEGs) using clusterProfiler R package[91]. We accessed C5 ontology collection from MSigDB (v7.4.1) using R package msigdbr, and only tested for Biological Process sets[58,66,67]. The proportion of DEGs in a gene set and FDR were employed to assess the GO pathway enrichment of DEGs.

## Cis-eQTL search

After QC, our eQTL analysis included 375 RNA-seq samples across 77 individuals. We tested variant–gene pairs for an eQTL effect if the variant was within 250 kb of the transcription start site (TSS) of the nearby gene. In total, we tested 7,475,856 SNP–gene pairs after removal of variants within the major histocompatibility complex (MHC) region of chromosome 6. The following model was used for the initial eQTL search:

$$E_{i,j} = \theta + \beta_{geno} \cdot g_j + (\kappa_i | j) + \sum_{l=1}^{20} \beta_{PC_l} \cdot PC_{i,l} + \sum_{m=1}^{3} \beta_{gPC_m} \cdot gPC_{j,m} + \sum_{n=1}^{10} \beta_n \cdot Site_{j,n}$$

(1)

where $E_{i,j}$ is the expression of gene for sample $i$ from the individual $j$. $\theta$ is an intercept term, and $\beta_{geno}$ is the effect (eQTL) of the genotype of individual $j$ ($g_j$). We included donor ($\kappa_i$) as a random effect, and 20 expression PCs ($\beta_{PC\,l}$) and 3 genotyping PCs ($\beta_{gPC\_m}$) and 10 recruitment sites as fixed effects. We chose 20 PCs to maximize the number of eQTL genes detected while minimizing the number of principal components we corrected for.

This model (1) was fit using the lmer() function from lme4[92].

To identify eQTL interactions, we tested the lead SNP for each eGene, fitting two models (2) and (3) for each SNP–gene pair tested[36]:

$$E_{i,j} = \theta + \beta_{geno} \cdot g_j + (\kappa_i | j) + \sum_{l=1}^{20} \beta_{PC_l} \cdot PC_{i,l} + \sum_{m=1}^{3} \beta_{gPC_m} \cdot gPC_{j,m}$$
$$+ \sum_{n=1}^{10} \beta_n \cdot Site_{j,n} + \beta_{var} \cdot x_i$$

(2)

$$E_{i,j} = \theta + \beta_{geno} \cdot g_j + (\kappa_i | j) + \sum_{l=1}^{20} \beta_{PC_l} \cdot PC_{i,l} + \sum_{m=1}^{3} \beta_{gPC_m} \cdot gPC_{j,m}$$
$$+ \sum_{n=1}^{10} \beta_n \cdot Site_{j,n} + \beta_{var} \cdot x_i + \beta_{int} \cdot x_i \cdot g_j$$

(3)

with additional terms for the effect of the variable ($x_i$) being tested ($\beta_{var}$) and interaction effect ($\beta_{int}$). We determined the significance of the interaction effect with a likelihood ratio test comparing these two models.

A significant interaction can either increase the original eQTL effect (magnifier) or decrease the eQTL effect (dampener) as the variable of interest ($x_i$) changes. In order to classify a eQTL interaction into magnifier or dampener, we multiplied the interaction $z$-score by the sign of the original eQTL effect ($\beta_{geno}$). The interactions with an adjusted $z$-score >0 are defined as magnifiers, and those with an adjusted $z$-score <0 are dampeners.

## IL-17 pathway score

Given the role of IL-17 in psoriasis disease activity, we calculated two types of IL-17 pathway score for each sample by summing the expression of IL-17 response genes. For the first, we used WikiPathways curated IL-17 signaling gene set that encompasses 32 genes from IL-17 cytokine family, their downstream receptors and response genes (*CEBPB, CEBPD, TRAF3IP2, IL17F, IL17RE, AKT1, IL17RA, IL17C, IL17B, GSK3B, IKBKB, IL17A, JAK1, JAK2, NFKB1, NFKBIB, PIK3CA, IL17D, IL17RD, IL17RB, MAPK1, MAPK3, RELA, IL25,* SP1, *STAT3, MAP3K7, TRAF3, TRAF6, IL17RC, IKBKG, MAP3K14*)[58,93].

We also calculated a keratinocyte-specific IL-17 pathway score with 23 IL-17 response genes (*IL19, SPRR2C, C15orf48, SLC6A14, S100A7A, DEFB4A, VNN3, CXCL8, IL6, CCL20, NES, S100A12, ALOX12B, SERPINB4, SERPINA3, CFB, RHCG, LCN2, SAA2, PDZK1IP1, IL36G, S100A7, IL17A*) selected from a previous study, in which Miura et al. stimulated human keratinocytes with IL-17A and profiled transcriptional responses to IL-17A stimulation[57]. Only differentially expressed genes with fold change >5 were included.

## GTEx tissue expression of interaction eGenes

To assess whether the SPITS-interaction eGenes are skin-expressed or immune-expressed, we obtained median gene-level TPM by tissue data from Genotype-Tissue Expression Portal (GTEx)[28]. We then compared expression of SPITS-interaction eGenes in whole blood and in skin (sun exposed and not sun exposed).

## Single-cell expression of interaction eGenes

To identify the specific cell type that a SPITS-interaction eGene is expressed in, we used data from a published single-cell study, which includes 141,626 cells from 3 psoriatic samples[32]. We re-grouped the pre-defined 41 cell states into 14 representative cell types, with seven types from the stromal compartment (keratinocytes, Schwann cells, lymphatic endothelium, melanocytes, pericytes, vascular endothelium, fibroblast), and the rest from the immune compartment (ILC_NK, T cells, plasma

cells, Langerhans cells, macrophages, dendritic cells, mast cells). We then compared mean expression of SPITS–interaction eGenes within each individual cell type with its global mean expression to determine its cell specificity.

## Motif enrichment analysis

We used the HOMER software suite[72] to look for enrichment of transcription factor binding motifs in the 116 SPITS-eQTL interactions (FDR < 0.2). We tested eQTL interactions for the lead SNP–the SNP with the strongest main effect–for that eQTL, but the lead SNP is not necessarily the functional SNP. Hence, we additionally considered all SNPs in the *cis* window with an $r^2 \geq 0.8$ with the lead SNP in the 1000 Genomes European population[44]. We defined our motif search window as 20 bp on either side of each SNP (i.e., 41 bp wide). To prevent false enrichment due to overcounting of intersecting windows, we merged them into non-overlapping windows prior to motif enrichment analysis. HOMER reported the transcription factor motifs that were significantly enriched in the sequences of interest, and the motifs were plotted using the SeqLogo R library[94].

## Cellular deconvolution

We used CIBERSORTx[65] to deconvolute cell fractions from the bulk RNA-seq data. We generate the signature matrix from the psoriatic single-cell data described above[32]. To avoid bias due to imbalanced counts of different cell types in the single-cell reference, we randomly sampled 1000 cells from each cell type; for the cell types that have fewer than 1000 cells, we included all of them in the reference. Since the reference profile used droplet-based single-cell sequencing, we applied S-mode batch correction as suggested[65]. To run CIBERSORTx, we set permutation to 500, and left all the other parameters as default.

## Forward selection

We performed forward stepwise logistic regression to determine the relative importance of the cell types in distinguishing SPITS of the samples. To fit the model, we used SPITS status as the outcome variable and predicted cell type proportions as predictor variables. We first built univariate logistic models that included only one cell type, and selected the cell type with chi-square $p$ value < 0.05 and the largest decrease in deviance compared to the null model. After that, we fit bivariate models conditioning on the selected variable, and again selected the model with largest decrease in deviance. The process continued until no variable had $p < 0.05$.

## Gene set enrichment analysis (GSEA)

We performed gene set enrichment analysis with the R package fgsea[68] using the gsea function. MSigDB gene sets (v7.4.1) were imported using the package msigdbr[58,66,67]. $p$-values were obtained from 10,000 permutations.

## Reporting summary

Further information on research design is available in the Nature Portfolio Reporting Summary linked to this article.

## Data availability

The raw gene expression data, sequencing data and summary statistics are deposited in dbGaP (accession code: phs003395). Access to dbGaP's data requires meeting NIH criteria, including holding a position equivalent to a tenure-track professor or senior scientist. To request access, a formal request has to be submitted, including research intent and policy adherence, for review by an institutional Signing Official and NIH Data Access Committee(s) (DAC). One can expect a response post DAC review, with timelines varying by dataset and process. The clinical data of PAUSE trial is available on the ITN TrialShare website (www.itntrialshare.org). GTEx eQTL (https://storage.googleapis.com/gtex_analysis_v7/single_tissue_eqtl_data/all_ snp_gene_associations/Skin_Not_Sun_Exposed_Suprapubic.allpairs.txt. gz) and median TPM data (https://storage.googleapis.com/gtex_ analysis_v8/rna_seq_data/GTEx_Analysis_2017-06-05_v8_RNASeQCv1.1. 9_gene_median_tpm.gct.gz) were obtained from GTEx portal (gtex-portal.org). The skin scRNAseq data was obtained from ArrayExpress (https://www.ebi.ac.uk/arrayexpress/experiments/E-MTAB-8142). GWAS summary statistics were downloaded from GWAS catalog (www. ebi.ac.uk/gwas), including psoriasis (GCST005527), systemic scleroderma (GCST009131) and eczema (GCST90044763).

## Code availability

Codes used for the analysis are publicly available at https://github. com/immunogenomics/PAUSE_eQTL.

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

## Acknowledgements

The PAUSE trial was conducted by the Immune Tolerance Network and S.R. is supported by the National Institute of Allergy and Infectious Diseases of the National Institutes of Health (UM1-AI-109565). Abatacept and abatacept placebo were provided by Bristol Myers Squibb.

## Author contributions

Q.X., J.M., K.I., A.N., N.L., and Y.B. conducted all of the statistical analyses under the supervision of S.R.. N.L., L.A.C., K.M.H., M.S.A., D.A.F., D.E.S., and J.G.K. helped to curate the clinical data, interpret the clinical data, and interpret the analyses. Q.X., J.M., and S.R. wrote the initial manuscript, and all of the authors contributed to the final manuscript.

## Competing interests

S.R. declares the following competing interests: S.R. serves as a founder for Mestag, Inc. and an scientific advisor for Pfizer, Jansen, Rheos, and Sonoma. S.R. also serves as a consultant for Abbvie and Sanofi. The other authors declare no competing interests.
