## [Peer Review File · Nature Communications]

Immunosuppression causes dynamic changes in expression QTLs in psoriatic skinEditorial Note: Parts of this Peer Review File have been redacted as indicated to remove third-party material where no permission to publish could be obtained.

REVIEWER COMMENTS

Reviewer #1 (Remarks to the Author):

Xiao, Mears et al utilised longitudinal skin biopsies to map cis-eQTLs in the context of psoriatic inflammation and colocalise eQTLs with susceptibility loci for psoriasis. Due to the small sample size of this cohort, authors used multiple samples for each individual and a linear mixed-effects model to increase power for eQTL mapping. Authors defined samples with SPITS status, a non-genetic factor associated with the transcriptional changes between lesional and non-lesional biopsies, then assessed for interaction between genotype and SPITS. Authors next performed cell type deconvolution on the bulk RNAseq data using single-cell sequencing data derived from a relevant study and provided evidence showing the eQTLs interacting inflammation status were mainly driven by the differences in cell abundance of dermal and epidermal states. This study demonstrates the potential of utilising context-interacting eQTLs to delineate disease susceptibility and treatment response, a topic of interest that has not been explored in depth. Small sample size and novelty are issues with this otherwise well-done work. Analytical structure and data presentation are similar to a reported study (Davenport et al. Genome Biol.2018; ref#34 in this manuscript). I also find the below-detailed concerns in the methods and descriptions of the analyses and results in this current version too extensive to merit publication in Nature Communications.

Major issues:

1. Authors identified 953 unique eGenes involving 24,374 SNP-gene pairs, and compared with GTEx cis-eQTLs in Fig.1C. To avoid overweighting the contribution of highly correlated SNPs, LD pruning is necessary. A more appropriate way to determine the shared or independent eQTLs while accounting for LD is to perform colocalisation analysis using a Bayesian approach e.g. as implemented in the R package coloc and its extensions such as moloc. This should also be used to compare the evidence for shared effects at eQTLs with psoriasis GWAS loci.
2. Authors need to justify the number of gene expression and population structure PCs used for the cis-eQTL mapping. Do these PCs fully capture the variations and maximise eQTL discovery? The same number of expression PCs was used for the interaction analysis thus I wonder whether this dampens the effects of interactions since the SPITS signature was also derived from gene transcription hence adjusting PCs in the model may mask the overall variability of this signature.
3. Authors used overly stringent Bonferroni cutoff for eQTL identification, which may result in substantial false negatives. It would be more appropriate to employ a permutation procedure to control for multiple testing and eGene identification, as implemented in FastQTL tool and the GTEx pipeline.
4. Macrophage shows differential abundance in SPITS-positive relative to SPITS-negative samples, which reassures its critical roles in mediating skin inflammation and wound healing. To what extent do these eQTLs interactions relate to macrophage specific effects? using cell proportion and/or 'Absolute Score' computed by CIBERSORTx. What is the trajectory of immune cell abundance upon ustekinumab and abatacept treatments?
5. The availability of the raw data, summary statistics and code underlying the results need to be carefully described. These data should be deposited in public repositories.

Minor issues:

1. The information about the ethnicities of the cohort are not mentioned. I recognise that the small

sample size would be a limitation factor, but it would be interesting to explore population-specific cis-eQTL signals.

2. How many of these significant eQTLs could be validated using an additive model at individual-level? What are the overlaps?

3. The population used for LD r^2 calculation should be specified in the text. For example in lines 166, 168, 373 etc.

4. The number of samples for differential expression analysis in Fig.S4a needs to be included in the legend. Was this done using data at individual level or data from all sample assigned with SPITS status?

5. In lines 201-202, it is confusing to report the result based on this cohort but cite GTEx paper in the text.

6. Authors need to provide evidence to support the results in lines 240-241.

7. In box plots showing the eQTL associations e.g. Fig.2b, instead of using genotype dose on the x axis, author should plot the ref/alt alleles.

8. Line 268-269, no difference in effect size was observed based on the text (-0.77 vs. -0.79).

9. Fig.2d was mislabelled as 'c'.

10. To support the results in line 229-233, correlation plots should be used in Supplementary Fig.S4c.

Reviewer #2 (Remarks to the Author):

The authors present an eQTL study using skin from 77 psoriasis patients. Although the sample size is small, and the authors acknowledge this weakness in their discussion, I still believe eQTL analysis on diseased tissue to be worthwhile and of interest to the readership. Please find my comments on this manuscript below:

1) The authors make extensive use of their skin psoriatic inflammation transcriptional score (SPITS), however the motivation for using this approach appears minimally stated. Please can the authors go into more detail describing the problem SPITS is trying to solve and how it achieves this. If I understand correctly, SPITS is helpful because lesional biopsies from treated patients can look more like non-lesional, however the authors also emphasize how controlled clinical trials help address factors such as medication differences, so explanation in this context would be helpful.

2) Please can the authors clarify the criteria for determining SPITS negative and SPITS positive? Currently they "considered samples that resembled baseline lesional samples", however this appears vague and unclear. More precise definitions would be welcome. Along these lines, how did the authors test whether using SPITS outperforms simply using the lesional / non-lesional labels? Did every individual have both a lesional and non-lesional sample at each time point that was used in the study? Perhaps it would be helpful to include a table outlining the samples, in addition to the current table outlining the individuals?

3) In the Methods section, the authors indicate they only used the top 2,942 genes used for PCA. Why was this? They also explain they included "20 gene expression PCs" but not sex or age, because the PCs were sufficient. Did the authors use any statistical techniques to determine the optimal number of

PCs and that sex/age did not have any further impact? I feel a little uncomfortable in preferring PCs to sex and age, since they are less easily interpretable. What would happen if the authors started with sex and age, and then added as many PCs as would have an impact beyond this? The authors also indicate they used recruitment site as a covariate - how was this coded?

4) In the Results, the authors claim the GTEx and psoriasis studies to have "similar eQTL effect sizes", but then only give one set of effect sizes and p-values (presumably for psoriasis?). Please would it be possible to give the effect sizes and p-values for both GTEx and psoriasis, so that the reader can see for themselves how similar they are? Also, since there is a high concordance with GTEx, what advantage does the psoriasis tissue study offer compared to just using GTEx?

5) Out of 63 risk loci for psoriasis, 5 are in high LD with an eSNP. However, as far as I can tell, the authors do not show how many loci the eSNPs come from? If it is more than 5, would this suggest some novel loci that might be of interest? This would be helpful, as overall the findings are not very novel, e.g. eGenes are more expressed in skin cells and LCE3C is the most important interaction.

6) Additional care should be taken to avoid confusion and clarify wording across the manuscript. For example, in the Introduction, the authors write that "psoriasis can affect the joints", and then describe psoriasis primarily in terms of skin tissue. I feel it would be better just to say that psoriasis is associated with arthritis. Do I understand correctly that the authors refer to eQTL interactions differently: 1) between genetics and gene expression; 2) for the impact of inflammatory status? If so, using the same term to mean two different things is confusing. Please can the authors go through the text to make sure all the wording is clear and unambiguous?

Minor points:

1) Some sentences are missing citations, for example "eQTL interactions can point to shared upstream regulatory mechanisms by prioritizing TFs with relevant binding motifs" and "hard to disentangle from the effect of drugs that the patients may be using".

2) When referring to the "375 samples from 77 genotyped patients" at the beginning of the Results, perhaps it would be helpful to refer to Supplementary Table 1?

3) Could pseudotime be a way to infer potential mechanisms in single cell analysis? If so, eQTLs are not necessarily the only way, and this should be clarified in the Introduction.

4) Figure 3a and 3c need labels for their color scale legend.

Reviewer #1 (Remarks to the Author):

Xiao, Mears et al utilised longitudinal skin biopsies to map cis-eQTLs in the context of psoriatic inflammation and colocalise eQTLs with susceptibility loci for psoriasis. Due to the small sample size of this cohort, authors used multiple samples for each individual and a linear mixed-effects model to increase power for eQTL mapping. Authors defined samples with SPITS status, a non-genetic factor associated with the transcriptional changes between lesional and non-lesional biopsies, then assessed for interaction between genotype and SPITS. Authors next performed cell type deconvolution on the bulk RNAseq data using single-cell sequencing data derived from a relevant study and provided evidence showing the eQTLs interacting inflammation status were mainly driven by the differences in cell abundance of dermal and epidermal states. This study demonstrates the potential of utilising context-interacting eQTLs to delineate disease susceptibility and treatment response, a topic of interest that has not been explored in depth. Small sample size and novelty are issues with this otherwise well-done work. Analytical structure and data presentation are similar to a reported study (Davenport et al. Genome Biol.2018; ref#34 in this manuscript). I also find the below-detailed concerns in the methods and descriptions of the analyses and results in this current version too extensive to merit publication in Nature Communications.

We thank the reviewer for their service and their feedback. We have tried to fully address their comments point-by-point as detailed below.

Major issues:

1. Authors identified 953 unique eGenes involving 24,374 SNP-gene pairs, and compared with GTEx cis-eQTLs in Fig.1C. To avoid overweighting the contribution of highly correlated SNPs, LD pruning is necessary. A more appropriate way to determine the shared or independent eQTLs while accounting for LD is to perform colocalisation analysis using a Bayesian approach e.g. as implemented in the R package coloc and its extensions such as moloc. This should also be used to compare the evidence for shared effects at eQTLs with psoriasis GWAS loci.

We thank the reviewer for raising these important and confusing issues. To be clear, in **Fig.1c** (which is now **Fig.1d** as we edited **Fig.1** per **reviewer 2's major issues #2**), we did not depict 24,374 significant SNP-gene pairs, since that analysis would indeed be confounded by linkage disequilibrium between nearby SNPs within the same locus acting as an eQTL on a single gene. Rather we only plotted the deduplicated SNP-gene pairs, both significant and non-significant in our study, including only the lead SNP for each gene that overlaps with significant GTEx SNP-gene pairs (N = 2,621). We have now made this clear in the main text and **Fig.1d** (previous **Fig.1c**) legend of the manuscript to avoid confusion:

Results:

Of 6,305,752 SNP-gene pairs tested, using a stringent Bonferroni p-value threshold ($p < 6.69e-9 = 0.05/6,305,752$), we reported 24,374 significant SNP-gene pairs (**Fig.1c**). With only the lead SNP for each gene, we identified 953 genes with at least one significant eQTL (eGenes) (**Supplementary Data 2**).

We examined significant eQTLs from GTEx and the pair of lead SNPs and eGenes for each of these eQTLs (N=2,621 eGenes). We observed a high degree of concordance between t-values of deduplicated SNP-gene pairs the PAUSE study and GTEx, suggesting our study identified highly concordant results with a study that is substantially larger in size (with 98.5% of effects in the same direction, Fig.1d, Supplementary Data 3).

Fig.1d legend:

(d) Concordance of ITN PAUSE deduplicated lead variant-gene pairs with significant eQTLs observed in the GTEx consortium²⁸ (FDR < 0.05) of 449 individuals. Each point represents a significant SNP-gene pair in this study. Concordant pairs are colored in black, discordant pairs are colored in grey.

As for the colocalization analysis, while we previously found 5 eSNPs that are in high LD ($LD > 0.7$) with reported psoriasis risk loci, we thank the reviewer for the suggestion and recognize that performing colocalization analysis using a Bayesian approach might be more appropriate. We used 'coloc' to assess overlap between our eQTLs and a psoriasis GWAS study^{19,43}. We removed **Supplementary Data 2** that contains only LD r^2 information and included a table **Supplementary Data 4-psoriasis** below that contains loci that either 1) have an $H4 > 0.75$ from eQTL and GWAS summary statistics, or 2) have an $r^2 > 0.5$ between eQTL lead SNPs and GWAS hits. In total, we reported 6 colocalizing loci with $H4 > 0.75$, and 5 lead eSNPs that have $r^2 > 0.5$ (**Supplementary Data 4-psoriasis**). Specifically, for loci LCE3C, IFNL1 and an RNA gene ENSG00000255389, we observed both high probability of colocalization and high LD with psoriasis GWAS hits. For LCE3C which we highlighted in the main text, we observed an $H4$ posterior probability of 0.996 with the psoriasis GWAS locus, showing strong evidence of the same variant influencing both the expression and trait. These results further increased our

confidence in the disease relevance of our eQTLs. We made updates to emphasize the colocalization results in **Supplementary Data 4-psoriasis** and in the main text:

Supplementary Data 4-psoriasis:

eQTL variant	Trait hits	r2	Locus	nSNPs	PP.H4	Symbol	Trait
rs1491377616	rs6677595	1.000	ENSG00000244057	149	9.961119e-01	LCE3C	Psoriasis
rs2927608	rs2910686	1.000	ENSG00000164308	427	3.136995e-06	ERAP2	Psoriasis
rs571694	rs1056198	0.979	ENSG00000158480	94	3.378441e-01	SPATA2	Psoriasis
rs59960858	rs7552167	0.961	ENSG00000185436	5	8.341801e-01	IFNLR1	Psoriasis
rs35815577	rs33980500	0.553	ENSG00000255389	133	9.785831e-01		Psoriasis
rs693824			ENSG00000172543	3	9.981350e-01	CTSW	Psoriasis
rs2231884			ENSG00000172803	4	9.980170e-01	SNX32	Psoriasis
rs11250130			ENSG00000104643	192	8.098779e-01	MTMR9	Psoriasis

Methods:

For each of the significant cis-eQTL genes, we performed colocalization analysis with publicly available GWAS studies using coloc (v.5.1)⁴³. We downloaded GWAS summary statistics of three diseases involving skin inflammation from GWAS Catalog (<https://www.ebi.ac.uk/gwas/>), including psoriasis (GCST005527), scleroderma (GCST009131) and eczema (GCST90044763). To detect colocalizing signals between eQTLs and GWAS variants, we used the posterior probability of sharing one common causal variant (i.e. PP.H4) > 0.75. To assess linkage disequilibrium (LD) between the previously reported psoriasis-associated variants and our eQTL lead SNPs, we calculated LD r2 between them, and identified the eSNPs that have high LD (r2 > 0.5) with psoriasis risk SNPs¹⁹.

Results:

Using coloc⁴³, we performed colocalization analysis to assess the probability of sharing one common causal variant (i.e., PP.H4) between PAUSE skin eGenes and psoriasis GWAS variants¹⁹. We also calculated linkage disequilibrium (LD) r2 between the lead eSNPs and psoriasis GWAS hits based on European population (EUR) in 1000 Genomes Project⁴⁴. We noted 6 of our eGenes colocalize (PP.H4 > 0.75) with psoriasis loci, and 5 lead eSNPs have r2 > 0.5 with psoriasis variants (Supplementary Data 4). For example, eSNPs influencing IFNLR1 expression colocalize with psoriasis variants (PP.H4 = 0.834) and the lead eSNP rs59960858 is in almost perfect LD (r2 = 0.961) with the psoriasis risk allele rs7552167 (Supplementary Data 4). IFNLR1 encodes a subunit of a cytokine receptor and has been shown to exert antiviral effect in the context of psoriasis skin barrier

breakdown⁴⁵. The gene has also been mapped to psoriasis risk alleles across different studies^{19,46–48}. Other colocalizing loci included eSNPs affecting expression of LCE3C, MTMR9, CTSW, SNX32, and RNA gene ENSG00000255389 (Supplementary Data 4).

As mentioned previously, we detected 6 colocalizations between our eQTL variants and psoriatic variants. Among them, we reported a posterior probability of 0.996 that skin eQTLs and psoriasis risk alleles share a same casual variant at the LCE3C locus. Furthermore, the eSNP rs1491377616 influencing LCE3C is in perfect LD ($r^2 = 1$) with the psoriasis-associated variant rs6677595 (Supplementary Data 4).

2. Authors need to justify the number of gene expression and population structure PCs used for the cis-eQTL mapping. Do these PCs fully capture the variations and maximise eQTL discovery? The same number of expression PCs was used for the interaction analysis thus I wonder whether this dampens the effects of interactions since the SPITS signature was also derived from gene transcription hence adjusting PCs in the model may mask the overall variability of this signature.

We thank the reviewer for raising these important points. Our initial choice of 20 PCs was based on the large proportion of variance captured (82.7% variance, **Supplementary Fig.1a**) and was consistent with other similar sized studies, where PCs were used to capture expression variation and maximize eQTL discovery. We have since conducted analyses to demonstrate results are not sensitive to the choice of 20 PCs. We performed a sensitivity analysis on chromosome 1 and chromosome 2 with a range of number of PCs (10, 15, 17, 19, 21, 22, 23, 25, and 30) in the model. From **Supplementary Fig.1b** below, we detected fewer significant eQTLs on chromosome 1 and 2 with 10 PCs. However, between 15-23 PCs, the number of significant eQTLs are largely similar. Given that the number of significant eQTLs was not sensitive to the choice of number of PCs, we retained our choice of 20 PCs. We have edited the **Methods** to include **Supplementary Fig PC Number**.

Methods:

We chose 20 PCs to maximize the number of eQTL genes detected while minimizing the number of principal components that explain a reasonable amount of variance (82.7% variance, **Supplementary Fig.1a**). We observed that the number of eQTLs identified was not very sensitive to the choice of $n=20$ PCs, with a similar number of eQTLs being identified using 15-25 PCs (**Supplementary Fig.1b**).

Supplementary Fig.1a:

Supplementary Fig.1b:

We next explored whether adjusting for gene expression PCs might dampen the effects of interactions. We have conducted an additional analysis to identify eQTL interactions with SPITS status without correcting for PC1 and PC2, which are significantly correlated with SPITS status (PC1-SPITS status $R = 0.64$, $p = 1.94e-45$; PC2-SPITS status $R = 0.66$, $p = 2.17e-47$). In this new analysis we detected fewer interactions (90 at $FDR < 0.05$, and 98 at $FDR < 0.20$) compared to the eQTL interaction analysis with SPITS status-correlated PCs (98 at $FDR < 0.05$, 116 at $FDR < 0.20$). The SPITS eQTL interaction betas were highly correlated ($R = 0.99$, $p < 1e-15$) between analyses where PC1 and 2 were excluded and included. Among the interactions that were significant in both analyses, excluding PC1

and PC2 on average nominally reduces the magnitude of the interaction terms by 1.8%. As PC1 and PC2 are also significantly correlated with other potential confounders like sex and race (PC1-sex $p = 1.71e-3$, PC2-sex $p = 8.91e-8$; PC1-race $p = 0.012$, PC2-race $p = 2.43e-6$), we decided to include them in the final model. We have added **Supplementary Fig.7b** and the following text to the results.

Results:

We also considered the possibility that the inclusion of expression PCs might be reducing the power to detect interactions. To further explore this, we repeated the SPITS status interaction analysis without correcting for principal components 1 and 2, which are significantly correlated with SPITS status (PC1-SPITS status $R = 0.64$, $p = 1.94e-45$; PC2-SPITS status $R = 0.66$, $p = 2.17e-47$). We found the betas for the interaction term are highly correlated ($R = 0.99$, $p < 1e-15$, **Supplementary Fig.7b**), and that including these PCs increased the number of eQTLs discovered (90 at $FDR < 0.05$ without PC1 and 2, versus 98 at $FDR < 0.05$ with PC1 and 2).

Supplementary Fig.7b:

3. Authors used overly stringent Bonferroni cutoff for eQTL identification, which may result in substantial false negatives. It would be more appropriate to employ a permutation procedure to control for multiple testing and eGene identification, as implemented in FastQTL tool and the GTEx pipeline.

We thank the reviewer for the valuable input. We didn't use FASTQTL as it employs a standard linear model, which doesn't allow us to include random effects. In the PAUSE

trial, since we can have multiple samples from one donor, it is critical to account for within-individual variability with a random effects term. Therefore, instead of applying FASTQTL, we used a linear mixed effects model with a stringent p-value which we felt would be appropriate. This approach may be overly conservative since for each gene tested, every SNP is not independent. While we might be losing power, our approach minimizes false positives. This conservative approach ensures that we were confident in the eQTL main effect before testing that effect for an interaction.

We agree with the reviewer that using Bonferroni correction might reduce the number of main effect eQTLs that we analyze, and hence the number of interactions we are detecting. To explore the effect of a less stringent cutoff, we relaxed our threshold from $6.7e-9$ to $1e-7$, which resulted in 1,266 significant eQTLs (previously 953). From these new significant eQTLs, we were able to find 145 SPITS interactions at $FDR < 0.20$ (previously 116), and 115 SPITS interactions at $FDR < 0.05$ (previously 98) (**Supplementary Fig.8a**). Compared to some of the previous interactions, the newly identified SPITS interactions have more modest eQTL effects and FDRs (**Supplementary Fig.8b**). We thus decided to stay with the more conservative threshold as we found the change in results was not very impressive even when we relaxed the threshold by more than 100-fold. We have added the following text to the discussion to address this point.

Results:

We considered whether a less stringent p-value threshold for original cis-eQTLs may have enabled us to identify more eQTL interactions. Arguably, a Bonferroni corrected $p < 6.69e-9$ threshold is too stringent since it does not account for LD within loci which would reduce the effective number of tests being conducted. To explore the effect of a more relaxed threshold, we ran a separate analysis using a lowered threshold ($p < 1e-7$). Further, we explored skin inflammation-eQTL interactions using a similarly relaxed threshold. Lowering the threshold by more than 100-fold led to the discovery of 1,266 significant eQTLs ($N = 953$ if Bonferroni p-value is applied), 145 SPITS interactions at $FDR < 0.20$ ($N = 116$ if Bonferroni p-value is applied) and 115 SPITS interactions at $FDR < 0.05$ ($N = 98$ if Bonferroni p-value is applied) (**Supplementary Fig.8a**). This relaxation resulted in only a moderate increase in signal with small effect sizes and larger FDRs (**Supplementary Fig.8b**). We therefore used the more conservative Bonferroni threshold to ensure that we were confident in the eQTL main effect before testing that effect for an interaction.

Discussion:

We acknowledge that our approach for eQTL discovery using a stringent Bonferroni corrected p-value threshold may be conservative and might reduce our ability to detect eQTLs with a modest effect. Consequently, this may reduce the number of interactions we observe. However, given the challenge of identifying interactions, we wanted to ensure that we were confident in the eQTL effect before testing that effect for an interaction.

Supplementary Fig.8a:

Supplementary Fig.8b:

4. Macrophage shows differential abundance in SPITS-positive relative to SPITS-negative samples, which reassures its critical roles in mediating skin inflammation and wound healing. To what extent do these eQTLs interactions relate to macrophage specific effects? Using cell proportion and/or 'Absolute Score' computed by CIBERSORTx. What is the

trajectory of immune cell abundance upon ustekinumab and abatacept treatments?

We thank the reviewer for raising the importance of macrophages in psoriasis. We ran the analysis with macrophage proportion derived from CIBERSORTx, and found 71 interactions at $FDR < 0.05$, and 82 interactions at $FDR < 0.20$. Among them, 69/71(97.2%) and 77/82(93.90%) were overlapping with SPITS interactions, respectively. Among the interaction eGenes, only one gene *SIGLEC12* was highly expressed in macrophages ($\log_{2}FC = 4.76$). Hence we want to be careful to emphasize that the interactions we found here might not necessarily be macrophage-specific, but may simply be associated with the presence of a higher proportion of macrophages, which is indicative of a higher level of inflammation. We have revised the manuscript to better reflect this interpretation and apologize for any confusion this may have caused.

Results:

We also found that the interacting eGenes were mostly expressed in the same differentially abundant cells (e.g. keratinocyte, fibroblast, melanocyte) between SPITS-positive and negative-samples (**Fig.3c**). We next assessed whether the eQTL interactions were also linked to these cell type proportions, which might be the result of a higher level of inflammation.

Results:

Given that macrophages play a crucial role in managing skin inflammation and promoting wound healing³, and that we reported higher proportion of macrophages in SPITS-positive samples compared to SPITS-negative, we further assessed interactions with macrophage proportion. We found 71 interactions at $FDR < 0.05$, and 82 interactions at $FDR < 0.20$. Among them, 69/71 (97.2%) and 77/82 (93.90%) were overlapping with SPITS interactions, respectively. Among the macrophage proportion-interacting eGenes, only one of them, *SIGLEC12*, is highly expressed in macrophages ($\log_{2}FC = 4.76$, Fig.3c). Many of these other eGenes may be occurring in non-macrophage cell-types; thus, eQTL interactions with macrophage proportion may be due to the proportion being a marker of inflammation level.

In terms of the trajectory of macrophage abundance upon treatments, we split the samples from first visit (before treatment) and after first visit (after treatment) and compared the inferred relative abundance of macrophages in SPITS positive negative and positive groups, respectively. We observed that after treatment, there is a significant decrease in macrophage abundance in both SPITS groups ($p\text{-value} = 6.39e-3$ in SPITS negative samples, $p\text{-value} < 1e-10$ in SPITS positive samples), while the change is more prominent in SPITS positive samples (mean fraction drops by $3.36e-4$ in SPITS negative samples, by $1.15e-2$ in SPITS positive samples). We included the following figure and text to illustrate the trajectory:

Results:

Given the crucial roles of macrophages in skin inflammation and wound healing³, we further explored the trajectory of macrophage proportions before and after treatment. To do that, we split the samples from first visit (before treatment) and after first visit (after treatment) and compared the inferred relative abundance of macrophages in SPITS positive negative and positive groups, respectively. We observed that after treatment, there is a significant decrease in macrophage abundance in both SPITS groups (p-value = 6.39e-3 in SPITS negative samples, p-value < 1e-10 in SPITS positive samples), while the change is more prominent in SPITS positive samples (mean fraction drops by 3.36e-4 in SPITS negative samples, by 1.15e-2 in SPITS positive samples, **Supplementary Fig.10**).

Supplementary Fig.10:

5. The availability of the raw data, summary statistics and code underlying the results need to be carefully described. These data should be deposited in public repositories.

We agree with the reviewer that the code and data should be made publicly available. For the code and summary statistics described in the manuscript, we have organized it in a github repository. We make reference to this github in the manuscript:

Code Availability:

Codes used for the analysis are publicly available at:
https://github.com/immunogenomics/PAUSE_eQTL

In addition, we will submit the raw genotype, RNA data and metadata to dbGAP, which we will make public upon acceptance of the manuscript. We added the **Data Availability** section in the manuscript.

Data Availability:

The raw gene expression and sequencing data have been submitted to dbGap (Accession number pending). The clinical data of PAUSE trial is available on the ITN TrialShare website (www.itntrialshare.org). GTEx eQTL and median TPM data were obtained from GTEx portal (gtexportal.org). The skin scRNAseq data was obtained from ArrayExpress (<https://www.ebi.ac.uk/arrayexpress/experiments/E-MTAB-8142>). The GWAS summary statistics were downloaded from GWAS catalog (www.ebi.ac.uk/gwas), including psoriasis (GCST005527), systemic sclerosis (GCST009131) and eczema (GCST90044763).

Minor issues:

1. The information about the ethnicities of the cohort are not mentioned. I recognize that the small sample size would be a limitation factor, but it would be interesting to explore population-specific cis-eQTL signals.

We thank the reviewer for the suggested analysis, and we agree that population-specific eQTLs would be worth investigating. However, in our study, 83% of the participants included for analysis are self-reported “White/Caucasian” (**Supplementary Table 1**), and 74.03% European ancestry based on genetic data (**Supplementary Fig.14**). Thus, we are underpowered to perform ancestry-specific analysis. We have now specifically stated this in the main text:

Discussion:

The relatively small size might limit power to detect cis-eQTL main effects and interaction effects, and the fact that the cohort is largely European ancestry limits our ability to examine population-specific eQTL signals (**Supplementary Fig.14**).

Supplementary Fig.14:

2. How many of these significant eQTLs could be validated using an additive model at individual-level? What are the overlaps?

We thank the reviewer for the suggested analysis. To validate the model at individual-level, we reran the eQTL analysis using only baseline lesional and non-lesional samples, and identified 575 significant eQTLs using the Bonferroni threshold ($p < 6.69e-9 = 0.05/7,475,856$). They are all overlapping with the eQTLs identified by all the samples, with some small differences in terms of lead eSNP. When plotting the effect sizes of both analyses, we found that they are highly concordant ($R = 1$, **Supplementary Fig.2a**). However, when comparing the t-values of both analyses, we found that individual-level analysis is less powerful in detecting the eQTL signals (**Supplementary Fig.2b**). We revised the manuscript to add this complementary analysis:

Results:

As a complementary analysis, we reran the analyses without accounting for multiple visits to see how the current model compares to a linear model using only the baseline samples ($N = 140$ samples versus 375 samples). For the first visit analysis we included 140 samples (generally one lesional and one non-lesional sample) from 74 individuals. For the analysis with all visits we included 375 samples from 77 individuals. As expected, linear mixed model including all the visits detected more eQTLs than the first time-point linear model ($N = 953$ vs. $N = 575$). While the effect sizes of the two models are highly concordant comparing significant eQTL pairs identified using the full model ($R = 1$), the t-values from the linear mixed model are consistently higher than the linear model (**Supplementary Fig.2a,b**), indicating that including repeat visits increases the power of detecting eQTL signals.

Supplementary Fig.2a:

Supplementary Fig.2b:

3. The population used for LD r^2 calculation should be specified in the text. For example in lines 166, 168, 373 etc.

We appreciate the reviewer bringing this to our attention. We have reworked the manuscript to include the information for LD r^2 calculation.

Results:

We also calculated linkage disequilibrium (LD) r^2 between the lead eSNPs and psoriasis GWAS hits based on European population (EUR) in 1000 Genomes Project⁴⁴.

For the input sequences, we included the eSNPs (FDR < 0.20) and the SNPs in high linkage disequilibrium (LD, $r^2 > 0.8$) within 500kB window around the TSS using the EUR reference panel, and then defined +/- 20bp intervals around these SNPs⁴⁴.

Moreover, the SNPs rs7442394 and rs2105749, which are both in high linkage disequilibrium ($r^2 = 0.91$ and $r^2 = 0.83$, respectively) in EUR population with the interaction eSNPs for UGT2B7 and CBR3, disrupt the ARE motifs⁴⁴ (Fig.4c).

4. The number of samples for differential expression analysis in Fig.S4a needs to be included in the legend. Was this done using data at individual level or data from all sample assigned with SPITS status?

We thank the reviewer for their valuable input on this aspect. The legend of **Supplementary Fig.4a** has been updated as follows:

Supplementary Fig.4a:

(a) Differentially expressed genes (DEGs) between SPITS positive and SPITS negative samples (N=375). Differential expression is defined by $FDR < 0.05$ and $|\log FC| > 1.5$. Upregulated genes are colored in red, downregulated genes are colored in blue, and non-significant genes are colored in grey.

The differential expression analysis was done using data from all samples assigned with SPITS status, we edited the main text to clarify this point:

Methods:

We used the R package edgeR to find the differentially expressed genes between SPITS-positive and -negative samples (N=375).

5. In lines 201-202, it is confusing to report the result based on this cohort but cite GTEx paper in the text.

We acknowledge the reviewer's comment on this point. The citation should be placed right after referring to the GTEx samples, which is prior to the result based on this cohort. We have amended the text accordingly.

6. Authors need to provide evidence to support the results in lines 240-241.

We appreciate the reviewer's identification of this problem. To support our claim in previous lines 240-241 about binary SPITS status outperforms continuous SPITS, we had performed interaction analysis separately using continuous SPITS, and detected 88 interactions at $FDR < 0.05$ (98 with binary SPITS status), and 93 interactions at $FDR < 0.20$ (116 with binary SPITS status). We have made changes to the manuscript as follows:

Results:

We also tested a continuous SPITS score, which detected fewer eQTL interactions (88 interactions at $FDR < 0.05$, 93 interactions at $FDR < 0.20$).

7. In box plots showing the eQTL associations e.g. Fig.2b, instead of using genotype dose on the x axis, author should plot the ref/alt alleles.

We are grateful for the reviewer's attention to this detail. The x axis of **Fig. 2b**, **Fig. 2c**, **Fig. 4b**, **Fig. 5a** and **Supplementary Fig. 1b** have been changed as suggested.

Fig.2b:

Fig.2c:

Fig.4b:

Fig.5a:

Supplementary Fig.1b:

8. Line 268-269, no difference in effect size was observed based on the text (-0.77 vs. -0.79).

We thank the reviewer for raising this issue, and we apologize for the confusion. -0.77 is the main effect beta without the interaction term, describing the significant association between rs2708954 genotype dose and expression of *IL37*. -0.79 is the interaction beta in the model with the interaction term, which describes the significant interaction between rs2708954-*IL37* eQTL and SPITS status. We re-edited the main text to avoid the confusion:

Results:

As an example of a SPITS-eQTL interaction, *IL37* expression is associated with rs2708954 (main effect beta = -0.77, $p = 3.73e-10$) and this negative effect is significantly magnified in inflamed skin samples (SPITS interaction beta = -0.79, $FDR = 7.73e-7$, Fig.2b). The beta in SPITS positive samples is the sum of the main effect and interaction effect from the model with interaction term (beta = -0.77 - 0.79 = -1.56).

9. Fig.2d was mislabelled as "c".

We thank the reviewer for noticing this error. The legend of Fig.2d has been changed accordingly.

Fig. 2d Legend:

(d) Median skin and whole blood expression of interacting eGenes from GTEx consortium. Points correspond to interaction eGenes.

10. To support the results in line 229-233, correlation plots should be used in Supplementary Fig.S4c.

We appreciate the reviewer's identification of this point. We added correlation plots between SPITS and IL17 gene scores to **Supplementary Fig. 4** as **Supplementary Fig. 4d**.

Supplementary Fig. 4d:

Reviewer #2 (Remarks to the Author):

The authors present an eQTL study using skin from 77 psoriasis patients. Although the sample size is small, and the authors acknowledge this weakness in their discussion, I still believe eQTL analysis on diseased tissue to be worthwhile and of interest to the readership. Please find my comments on this manuscript below:

We thank the reviewer for their service, and for recognizing the value of our manuscript.

1) The authors make extensive use of their skin psoriatic inflammation transcriptional score (SPITS), however the motivation for using this approach appears minimally stated.

Please can the authors go into more detail describing the problem SPITS is trying to solve and how it achieves this. If I understand correctly, SPITS is helpful because lesional biopsies from treated patients can look more like non-lesional, however the authors also emphasize how controlled clinical trials help address factors such as medication differences, so explanation in this context would be helpful.

We thank the reviewer for encouraging us to clearly state the motivation for using SPITS. In the PAUSE trial, skin biopsies were collected on scheduled clinical visits. We wanted to clarify that lesional and non-lesional status were only determined at baseline; after the initial assessment, biopsies were obtained from the same spot in the skin, and whether the sample was classified as lesional or non-lesional was based on the appearance of the skin at baseline. Therefore, the lesional status provided by the clinicians only indicates the presence of inflammation in the skin at the first visit and doesn't necessarily reflect its status at the time of the biopsy, and does not necessarily reflect whether disease active or quiescent. Over time and with treatment, the inflammation may abate. Below are photos of a sample lesion taken from a past psoriasis clinical study (Zhu et al., 2018), where **a** and **b** panels presented lesions prior to treatment, and panels **c** and **d** showed the same lesional regions after treatment, illustrating the resolving process of a psoriatic lesion. Thus, in our study, although a lesional skin was marked as lesional from the first visit, the biopsy taken from a later visit could exhibit less redness and patchiness, as the same skin position has changed and resolved after treatments. Therefore, we applied SPITS to evaluate the inflammation level of a skin biopsy by leveraging its transcriptional profile. We believe that biopsies undergoing active inflammation should have higher inflammation signatures that could be captured by gene expression level. We have made changes to the manuscript to clarify the motivation of using SPITS in Results.

[redacted]

Results:

However, in the PAUSE trial, lesional and non-lesional status were determined on the first visit; future lesional or non-lesional biopsies were taken from the same site based on its appearance at the first visit. Therefore, the lesional status provided by the clinicians only indicates the presence of skin inflammation at the first visit and doesn't necessarily reflect its status at subsequent visits, where local inflammation

may have abated after treatment. In this trial, lesional status of the skin biopsies are subject to change, as participants showed clinical improvement in lesional skin following treatment with ustekinumab^{11,12}. PCA of samples reveals that at baseline the lesional and non-lesional samples separate along PC1 and PC2; however, at subsequent timepoints following ustekinumab treatment, the baseline lesional samples were less distinguishable from baseline non-lesional samples along the same PCs (**Supplementary Fig.4a**).

We therefore defined a skin psoriatic inflammation transcriptional score (SPITS) based on lesional and non-lesional biopsies at baseline, recognizing that biopsies undergoing active inflammation should have higher inflammation signatures that could be captured by gene expression level.

2) Please can the authors clarify the criteria for determining SPITS negative and SPITS positive? Currently they "considered samples that resembled baseline lesional samples", however this appears vague and unclear. More precise definitions would be welcome. Along these lines, how did the authors test whether using SPITS outperforms simply using the lesional / non-lesional labels? Did every individual have both a lesional and non-lesional sample at each time point that was used in the study? Perhaps it would be helpful to include a table outlining the samples, in addition to the current table outlining the individuals?

We thank the reviewer for their suggestion on this aspect and we apologize for the confusion. SPITS is a continuous score intended to capture the range of inflamed to uninfamed phenotypes present in lesional and non-lesional psoriatic skin, respectively. It is calculated as the centered linear discriminate score from an LDA classifier trained on lesional vs. non-lesional biopsies at the initial baseline timepoint, when they were assessed by clinicians. Assuming lesional skin represents the inflamed end of the spectrum and non-lesional skin represents the uninfamed end of the spectrum, the resulting SPITSs span this axis (**Supplementary Fig.4a**). We have amended the **Methods** and **Results** section to clarify the classification:

Methods:

SPITS is a continuous score capturing the range of inflammation observed across lesional and non-lesional samples; samples with positive SPITS values have the highest level of inflammation (characteristic of lesional samples at baseline), and the samples with negative values have low inflammation (characteristic of non-lesional samples at baseline). We denoted samples with positive SPITS to be SPITS positive, and samples with negative SPITS to be SPITS negative.

Results:

We applied linear discriminant analysis (LDA), using the skin transcriptional data from the first visit (baseline) as training set (N =140); these samples had their status determined by a clinician (Methods). We applied this classifier to post-first visit samples, which were unlabeled (N = 235) (Methods). We calculated a SPITS

for each sample based on the first 48 RNA-seq PCs (>90% variance explained) (Supplementary Fig.4b). Positive SPITS corresponded to lesional-like (i.e., inflamed) samples, while negative SPITS corresponded to non-lesional-like (i.e., less inflamed) samples — which presumably includes non-lesional and resolving lesional samples. Based on 10-fold cross-validation, we observed that SPITS was 95.00% (s.d. = 4.82%) accurate at separating the baseline lesional and non-lesional skin samples (Supplementary Fig.4c).

SPITS outperforms lesional/non-lesional labels because it accounts for the dynamic nature of skin inflammation in a systematic way. While SPITS uses transcriptional profile to infer inflammatory status of a sample, lesional/non-lesional status was defined only at the baseline and would not accurately represent lesional samples that resolved over time (**Fig. 3a**). When it comes to eQTL interaction analysis, lesional/non-lesional labels generated 39 eQTL interactions at $FDR < 0.5$, and 42 eQTL interactions at $FDR < 0.2$, while SPITS interactions are higher in number (98 at $FDR < 0.5$, 116 at $FDR < 0.2$). We amended the main text to emphasize this point.

Results:

We also tested a continuous SPITS score, which detected fewer eQTL interactions (88 interactions at $FDR < 0.05$, 93 interactions at $FDR < 0.20$). An alternative strategy testing interactions with lesional versus non-lesional labels as assigned at baseline resulted in even fewer eQTLs (39 eQTL interactions at $FDR < 0.5$, 42 eQTL interactions at $FDR < 0.2$).

To better outline the sample information in addition to the individual-level information, we included a **Fig.1d** and a separate table (**Supplementary Data 1**). From the heatmap, we can see that almost all subjects (94.81%) have at least one lesional and non-lesional sample.

Figure 1d:

3) In the Methods section, the authors indicate they only used the top 2,942 genes used for PCA. Why was this? They also explain they included "20 gene expression PCs" but not sex or age, because the PCs were sufficient. Did the authors use any statistical techniques to determine the optimal number of PCs and that sex/age did not have any further impact? I feel a little uncomfortable in preferring PCs to sex and age, since they are less easily interpretable. What would happen if the authors started with sex and age, and then added as many PCs as would have an impact beyond this? The authors also indicate they used recruitment site as a covariate - how was this coded?

We thank the reviewer for raising the important points. We used the top 2,942 genes for running PCA because they are the genes in the RNA-seq data with mean expression and standard deviation > 70% of all genes, suggesting they are among the most variable genes (**Methods**). We agree with the reviewer on the importance of including potential confounders like sex or age in the model. We favor PCs as they do not limit us to the confounders that have been measured in the study, but also capture confounding effects that are not explicitly captured but have global effects on expression. However, we recognize the importance of assessing that the PCs are capturing the effects of known confounders too. For more detailed analysis, please see our response to **Reviewer 1's Major Issue #2**.

To address the reviewer's comments, we have re-run the interaction analysis including sex and age in addition to the PCs. And found they have very little effect on the interaction betas ($R=1$, $p < 1e-15$, average change in nominal betas = 1.4%), suggesting the principal components are capturing these known confounders (**Supplementary Fig.7a**). We have included the following text in the manuscript to address this point.

Methods:

The 2,942 genes with mean expression and standard deviation > 70% of all genes (i.e. top 30% most variable genes) were included to conduct principal component (PC) analysis using the `prcomp` function from the stats R package.

Results:

To evaluate whether including PCs would be sufficient for correcting potential confounding factors such as age or sex, we re-ran the interaction analysis including age and sex as fixed effects. We found that including age and sex in addition to PCs has almost no effect on the interaction betas ($R=1$, $p < 1e-15$, average change in nominal betas = 1.4%), suggesting the principal components are capturing these known confounders (**Supplementary Fig.7a**).

Supplementary Fig.7a:

As for including recruitment site as a covariate, we thank the reviewer for pointing this out. We actually ran the analysis with recruitment site as fixed effect, but inadvertently failed to include it in the description of the models in the **Methods** section. We have adjusted the following text and models in the manuscript to address this error.

Methods:

Study Design

The purpose of this study was to identify eQTLs in a cohort of individuals with psoriasis participating in the ITN PAUSE clinical trial. The trial was described in detail by Harris et al. in the previous clinical paper³⁷.

Cis-eQTL Search

$$E_{i,j} = \theta + \beta_{geno} \cdot g_j + (\kappa_i|j) + \sum_{l=1}^{20} \beta_{PC_l} \cdot PC_{i,l} + \sum_{m=1}^3 \beta_{gPC_m} \cdot gPC_{j,m} + \sum_{n=1}^{10} \beta_n \cdot Site_{j,n}$$

Where $E_{i,j}$ is the expression of gene for sample i from the individual j . θ is an intercept term, and β_{geno} is the effect (eQTL) of the genotype of individual j (g_j). We included donor (κ_i) as a random effect, 20 expression PCs (PC 1), 3 genotyping PCs (gPC_m) and 10 recruitment sites as fixed effects.

$$E_{i,j} = \theta + \beta_{geno} \cdot g_j + (\kappa_i|j) + \sum_{l=1}^{20} \beta_{PC_l} \cdot PC_{i,l} + \sum_{m=1}^3 \beta_{gPC_m} \cdot gPC_{j,m} + \sum_{n=1}^{10} \beta_n \cdot Site_{j,n} \\ + \beta_{var} \cdot x_i$$

$$E_{i,j} = \theta + \beta_{geno} \cdot g_j + (\kappa_i|j) + \sum_{l=1}^{20} \beta_{PC_l} \cdot PC_{i,l} + \sum_{m=1}^3 \beta_{gPC_m} \cdot gPC_{j,m} + \sum_{n=1}^{10} \beta_n \cdot Site_{j,n} \\ + \beta_{var} \cdot x_i + \beta_{int} \cdot x_i \cdot g_j$$

4) In the Results, the authors claim the GTEx and psoriasis studies to have "similar eQTL effect sizes", but then only give one set of effect sizes and p-values (presumably for psoriasis?). Please would it be possible to give the effect sizes and p-values for both GTEx and psoriasis, so that the

reader can see for themselves how similar they are? Also, since there is a high concordance with GTEx, what advantage does the psoriasis tissue study offer compared to just using GTEx?

We are grateful for the reviewer's insight on this matter. In **Fig.1c**, we compared the effect sizes and p-values between psoriatic skin samples from PAUSE trial and healthy skin samples from GTEx study, and found high concordance between the significant eQTLs, which suggests that despite smaller in size, our study was able to identify comparable results with substantially larger studies like GTEx. We included a **Supplementary Data 3** with main effects, p-values of overlapping variant-gene pairs. Despite high concordance, GTEx didn't recruit individuals with dermatologic diseases, or with psoriasis. Whereas in PAUSE trial, by leveraging psoriatic skin samples and the clinical design, which were not available in GTEx, we are able to identify those eQTLs whose effect sizes are modulated by skin inflammation. We include this text in the manuscript to highlight the point.

Results:

To confirm that our eQTLs were consistent with prior eQTLs reported in the skin, we compared significant SNP-gene pairs from this dataset to cis-eQTLs reported by the GTEx consortium among 517 healthy skin samples from 449 individuals who were not selected for a particular condition²⁸. We examined significant eQTLs from GTEx and the pair of lead SNPs and eGenes for each of these eQTLs (N=2,621 eGenes). We observed a high degree of concordance between t-values of deduplicated SNP-gene pairs the PAUSE study and GTEx, suggesting our study identified highly concordant results with a study that is substantially larger in size (with 98.5% of effects in the same direction, **Fig.1d, Supplementary Data 3**).

5) Out of 63 risk loci for psoriasis, 5 are in high LD with an eSNP. However, as far as I can tell, the authors do not show how many loci the eSNPs come from? If it is more than 5, would this suggest some novel loci that might be of interest? This would be helpful, as overall the findings are not very novel, e.g. eGenes are more expressed in skin cells and LCE3C is the most important interaction.

We thank the reviewer for noticing this. For the 5 eSNPs that are in high LD with psoriasis risk loci, they are from 5 different eGenes. Please see **Reviewer #1, Response #1**, where we extended our colocalization analysis using coloc⁴³, and identified more loci that are colocalizing with psoriasis GWAS variants beyond LCE3C (PP.HH4 > 0.75).

Furthermore, to introduce more novelty to our current study, we performed a colocalization analysis of psoriatic skin eQTLs with summary statistics from GWAS of other inflammatory skin diseases, including eczema and systemic sclerosis (**Supplementary Data 4-eczema, Supplementary Data 4-scleroderma**). This comparative approach allowed us to identify potential shared causal variants between psoriatic skin eQTLs and other skin diseases. By examining colocalization of eQTLs

across different skin diseases, we can gain a more comprehensive understanding of the genetic basis of these conditions and potentially identify common pathways that can be targeted for therapeutic interventions. We edited the manuscript with the updated colocalization analysis, and we addressed the reviewer's comment by quantifying the number of loci with colocalization by including the supplementary tables and updating the **Results** section.

Supplementary Data 4-eczema:

eQTL variant	Trait hits	r2	Locus	nSNPs	PP.H4	Symbol	Trait
rs3006451			ENSG00000163218	283	0.9890073	PGLYRP4	Eczema
rs7515928			ENSG00000117151	396	0.8103702	CTBS	Eczema
rs11553576			ENSG00000149499	177	0.9307656	EML3	Eczema
rs202091352			ENSG00000135541	79	0.9108796	AH11	Eczema
rs1972660076			ENSG00000234465	240	0.9267709	PINLYP	Eczema
rs705699			ENSG00000139531	50	0.9880222	SUOX	Eczema
rs10876864			ENSG00000197728	48	0.9738071	RPS26	Eczema

Supplementary Data 4-scleroderma:

eQTL variant	Trait hits	r2	Locus	nSNPs	PP.H4	Symbol	Trait
rs1589845305	rs6598008	0.783	ENSG00000247095	182	0.0006468152	MIR210HG	Scleroderma
rs2058778859			ENSG00000168291	632	0.9916272370	PDHB	Scleroderma
rs4313195			ENSG00000071894	35	0.9207955953	CPSF1	Scleroderma
rs12684934			ENSG00000226752	234	0.785332647	CUTALP	Scleroderma

Results:

To investigate the functional impact of psoriatic skin cis-eQTLs in inflammatory skin diseases more broadly, we expanded our analysis to include eczema and systemic scleroderma risk loci. We extracted GWAS summary statistics for the two additional traits and performed colocalization analysis of the identified cis-eQTLs and disease GWAS loci to identify potential shared causal variants^{49,50}. Our results revealed 10 colocalizing loci, including 7 eSNPs colocalizing with eczema risk loci, and 3 colocalizing with scleroderma risk loci (**Supplementary Data 4**). The eGene that most strongly colocalized with eczema risk alleles was *PGLYRP4* (PP.H4 = 0.989) (**Supplementary Data 4**), which is an innate immunity-related gene encoding for peptidoglycan recognition protein. The deficiency of *Pglyrp4* in mice has been reported to be involved in the development of eczema through reduced recruitment of Tregs and increased activation of Th17 responses⁵¹. In addition, eSNPs affecting *PDHB* expression colocalize with scleroderma risk

alleles (PP.H4 = 0.992; **Supplementary Data 4**). *PDHB* encodes for a subunit of pyruvate dehydrogenase complex, which plays an essential role in metabolism and has been associated with cancer and neurological diseases^{52,53}. Additionally, a study conducted in mice showed that pyruvate dehydrogenase deficiency (PDH) can lead to metabolic shift in keratinocytes which may in turn result in loss of epidermal stem cells⁵⁴. The overlap between psoriatic cis-eQTLs and risk loci for other skin conditions provides another way of confirming and exploring the important inflammation related signal present in our data.

Discussion:

We also performed colocalization analysis between psoriatic skin eQTLs and GWAS of other inflammatory skin diseases, like eczema and scleroderma. This comparative approach allowed us to identify potential shared causal variants between psoriatic skin eQTLs and other skin diseases. For instance, we found that psoriatic skin eSNPs and eczema risk loci colocalize at *PGLYRP4*, which has been linked to eczema development in mouse models through Th17 activation⁵¹. Similarly, we observed colocalization at the *PDHB* gene with scleroderma. The gene encodes for a subunit of pyruvate dehydrogenase, and the PDH enzyme was reported to be associated with skin homeostasis in mice⁵². By examining colocalization of eQTLs across different skin diseases, we may gain a more comprehensive understanding of the genetic basis of these conditions and potentially identify common pathways that underlie these conditions or that can be targeted for therapeutic interventions. Our skin eQTL dataset thus provides a valuable resource for identifying potentially causal variants that are involved in the pathogenesis of inflammatory skin diseases.

6) Additional care should be taken to avoid confusion and clarify wording across the manuscript. For example, in the Introduction, the authors write that "psoriasis can affect the joints", and then describe psoriasis primarily in terms of skin tissue. I feel it would be better just to say that psoriasis is associated with arthritis. Do I understand correctly that the authors refer to eQTL interactions differently: 1) between genetics and gene expression; 2) for the impact of inflammatory status? If so, using the same term to mean two different things is confusing. Please can the authors go through the text to make sure all the wording is clear and unambiguous?

We thank the reviewer for bringing this to our attention. We have modified the manuscript as follows:

Introduction:

Psoriasis can cause thickened red skin lesions characterized by keratinocyte hyperproliferation, angiogenesis, and immune cell infiltration^{3,4}. In about 30% of cases, psoriasis is associated with an inflammatory arthritis⁵.

In the manuscript, when we mention eQTL interactions, we specifically refer to the latter point that the reviewer proposed (i.e. eQTL effect under the impact of environmental factors, like inflammatory status). These eQTL interactions are a special case of eQTLs, as their effects depend on environmental factors, and they are distinct from interactions between genetics and gene expression. To avoid confusion, we have reworked the introduction as below:

Introduction:

For instance, an environmental factor may alter regulatory factors that bind the eSNP, resulting in a genotype-by-environment statistical interaction, where the eQTL effect is amplified or dampened by the presence of the environmental factor²⁴⁻²⁹. In the context of psoriasis, inflammatory status of the skin may alter regulatory factors that bind the eSNP, leading to different degrees of eQTL gene expression changes upon the same eQTL. Here, eQTL interactions may serve as a proxy for regulatory changes in the immune or dermal and epidermal cell states when the status of the skin shifts from its non-lesional state to a psoriatic lesion.

Minor points:

1) Some sentences are missing citations, for example "eQTL interactions can point to shared upstream regulatory mechanisms by prioritizing TFs with relevant binding motifs" and "hard to disentangle from the effect of drugs that the patients may be using".

We appreciate the reviewer bringing this to our attention. We have edited the manuscript to include the references for the listed sentences.

2) When referring to the "375 samples from 77 genotyped patients" at the beginning of the Results, perhaps it would be helpful to refer to Supplementary Table 1?

We thank the reviewer for the suggestion. We agree that referring to **Supplementary Table 1** would be helpful. We modified the beginning of the Results section accordingly.

Results:

After stringent quality control, we had RNA-seq data on 375 samples from 77 genotyped patients (**Fig.1a,b, Supplementary Table 1, Supplementary Data 1**).

3) Could pseudotime be a way to infer potential mechanisms in single cell analysis? If so, eQTLs are not necessarily the only way, and this should be clarified in the Introduction.

We thank the reviewer for raising the possible application of pseudotime analysis in single-cell data. We appreciate the power of pseudotime analysis of inferring the

progression of individual cells along a biological process such as disease progression. The analysis allows us to identify differentially expressed genes along a single-cell pseudotime trajectory and can provide us useful insights into key regulatory events driving cell state transitions. However, compared to eQTL analysis, it has limited ability to inform us about changes in the effects of gene regulatory variants. We updated the **Introduction** section to include the potential use of pseudotime analysis.

Introduction:

Just as single-cell data may be a useful strategy to capture differences in cell states; analytical approaches such as pseudotime analysis and differential abundance analysis³³⁻³⁵ help infer the progression of cells through biological processes. However, these approaches may not reveal systematic changes in gene regulation and the potentially pathogenic variants mediating these changes.

4) Figure 3a and 3c need labels for their color scale legend.

We are grateful for the reviewer's attention to the detail. The color scale legend for **Figure 3a** and **3c** were added accordingly. Below is the modified **Figure 3**.

Figure 3:

REVIEWERS' COMMENTS

Reviewer #1 (Remarks to the Author):

The authors have addressed most of my previous concerns. Before publication, I have an additional comment about the coloc analysis:

The effect allele associated with disease risk is often the minor allele, and therefore, in some cases, the reference allele can be the effect/risk allele. In the revised version, authors matched the effect alleles of GWAS to the altered alleles of eQTL, which may result in substantial number of SNPs missed for the coloc analysis. Authors need to check the GWAS/eQTL allele codings before doing the merge and coloc.

Reviewer #2 (Remarks to the Author):

I am grateful for the time and effort the authors have invested to address my concerns, which I believe they have done well. However, I still feel it should be made clearer (in the Methods) that positive/negative SPITS means above/below the LDA line. Other than that I have no further changes to suggest.

Reviewer #1 (Remarks to the Author):

The authors have addressed most of my previous concerns. Before publication, I have an additional comment about the coloc analysis:

The effect allele associated with disease risk is often the minor allele, and therefore, in some cases, the reference allele can be the effect/risk allele. In the revised version, authors matched the effect alleles of GWAS to the altered alleles of eQTL, which may result in substantial number of SNPs missed for the coloc analysis. Authors need to check the GWAS/eQTL allele codings before doing the merge and coloc.

We thank the reviewer for bringing this to our attention. In the revised colocalization analysis, when we pair up the GWAS and eQTL data, we indeed matched the GWAS SNPs based on their effect/risk alleles to both reference and altered alleles of eQTLs to prevent losing SNPs for the coloc analysis. We have now amended the **Methods** section to avoid confusion.

Methods:

To detect colocalizing signals between eQTLs and GWAS variants, we matched the GWAS SNPs based on their effect/risk alleles to both reference and altered alleles of eQTLs, and used the posterior probability of sharing one common causal variant (i.e. $PP.H4 > 0.75$). We noted that for all three diseases we had p-values for each SNP. For scleroderma and eczema GWAS studies the effect coefficients were not publicly available. So for the coloc analysis, we calculated the probability of colocalization using coefficients and variance for psoriasis, and p-values for eczema and scleroderma.

Reviewer #2 (Remarks to the Author):

I am grateful for the time and effort the authors have invested to address my concerns, which I believe they have done well. However, I still feel it should be made clearer (in the Methods) that positive/negative SPITS means above/below the LDA line. Other than that I have no further changes to suggest.

We thank the reviewer for their suggestion. We have modified the **Methods** section to clarify the definition SPITS.

Methods:

SPITS is a continuous score capturing the range of inflammation observed across lesional and non-lesional samples. It is defined by centered linear discriminant score from linear discriminant analysis (LDA); samples with positive SPITS values (i.e. positive linear discriminant score, above LDA line) have the highest level of inflammation (characteristic of lesional samples at baseline), and the samples with negative values (i.e.

negative linear discriminant score, below LDA line) have low inflammation (characteristic of non-lesional samples at baseline).